# TALPID3 and ANKRD26 selectively orchestrate FBF1 localization and cilia gating

Hao Yan[1,2,3,7], Chuan Chen[4,7], Huicheng Chen[1,2,3], Hui Hong[1], Yan Huang[4,5], Kun Ling[4,5], Jinghua Hu[4,5,6✉] & Qing Wei[3✉]

Transition fibers (TFs) regulate cilia gating and make the primary cilium a distinct functional entity. However, molecular insights into the biogenesis of a functional cilia gate remain elusive. In a forward genetic screen in *Caenorhabditis elegans*, we uncover that TALP-3, a homolog of the Joubert syndrome protein TALPID3, is a TF-associated component. Genetic analysis reveals that TALP-3 coordinates with ANKR-26, the homolog of ANKRD26, to orchestrate proper cilia gating. Mechanistically, TALP-3 and ANKR-26 form a complex with key gating component DYF-19, the homolog of FBF1. Co-depletion of TALP-3 and ANKR-26 specifically impairs the recruitment of DYF-19 to TFs. Interestingly, in mammalian cells, TALPID3 and ANKRD26 also play a conserved role in coordinating the recruitment of FBF1 to TFs. We thus report a conserved protein module that specifically regulates the functional component of the ciliary gate and suggest a correlation between defective gating and ciliopathy pathogenesis.

[1] CAS Key Laboratory of Insect Developmental and Evolutionary Biology, CAS Center for Excellence in Molecular Plant Sciences, Institute of Plant Physiology and Ecology, Chinese Academy of Sciences, Shanghai 200032, China. [2] University of Chinese Academy of Sciences, Beijing 100039, China. [3] Center for Reproduction and Health Development, Institute of Biomedicine and Biotechnology, Shenzhen Institutes of Advanced Technology, Chinese Academy of Sciences (CAS), Shenzhen 518055, China. [4] Department of Biochemistry and Molecular Biology, Mayo Clinic, Rochester, MN 55905, USA. [5] Mayo Translational PKD Center, Mayo Clinic, Rochester, MN 55905, USA. [6] Division of Nephrology and Hypertension, Mayo Clinic, Rochester, MN 55905, USA. [7] These authors contributed equally: Hao Yan, Chuan Chen. ✉email: hu.jinghua@mayo.edu; qing.wei@siat.ac.cn

Cilia are microtubule-based subcellular organelles that arise from the basal body (derived from the mother centriole) and protrude from the surfaces of most eukaryotic cells[1,2]. Intraflagellar transport (IFT) machinery, which consists of IFT-A and IFT-B subcomplexes and mediates the bidirectional movement of IFT cargos along the axoneme, is required for the biogenesis and maintenance of all cilia across species[3,4]. Based on their motility, cilia are divided into motile cilia and immotile cilia (also called primary cilia). Motile cilia function as motile devices to propel cells or generate flow across cell surfaces[5]. Primary cilia act as cellular "antennae", and participate in responses to various environmental stimuli (thermal, mechanical, or chemical) and various signal transduction pathways critical for the normal development and homeostasis of organs, including the sonic Hedgehog (Shh), Wnt, and various G-protein-coupled receptor signaling pathways[6–10]. Consistent with the ubiquitous presence of the cilia in most cell types in the human body, cilia dysfunctions cause dozens of syndromic disorders, such as Joubert syndrome (JBTS), Meckel syndrome (MKS), nephronophthisis (NPHP), and Bardet–Biedl syndrome (BBS), collectively termed ciliopathies[11,12].

As a unique cellular organelle that is not membrane-enclosed, like the Golgi or lysosome, the primary cilium needs a stringent gating mechanism at the ciliary base to control the trafficking of various membrane and soluble proteins between the cytoplasm and ciliary compartment[13–16]. Transition fibers (TFs), which originate from transformation of the distal appendages (DAs) of the mother centriole during ciliogenesis, anchor the basal body to the membrane and represent the boundary between the apical membrane and the ciliary membrane[2,17]. TFs, together with the adjacent transition zone (TZ, the proximal part of the axoneme that contains highly organized Y-links), have been suggested to be key subdomains of the proposed ciliary gate[18,19]. Consistent with the proposed importance of TFs in the context of cilia, mutations in TF structural components CEP164, SCLT1, and CEP83 or TF-related proteins OFD1 and C2CD3 have been linked to various ciliopathies[20–24]. We previously reported that FBF1 specifically localizes to TFs, where it acts as a functional component of the ciliary gate[25]. The fact that among the first six TF components (CEP164, CEP83, CEP89, SCLT1, FBF1, and LRRC45) identified[25–32], FBF1 is the only one conserved from C. elegans to humans suggests that the FBF1 pathway likely represents a central part of TF-regulated cilia gating. Thus, understanding how the FBF1 pathway is regulated will reveal more key players of the ciliary gate.

Human KIAA0586, also known as TALPID3, is a ciliopathy protein. The TALPID3 gene was originally identified in an embryonic lethal mutant of chicken with defective Shh signaling[33–35]. Subsequent studies revealed that TALPID3 encodes a conserved centriole distal-end protein and that its roles in ciliogenesis are conserved across vertebrates, including zebrafish, mice, and humans[36–38]. Recently, mutations in TALPID3 were found to cause the ciliopathy JBTS[39–43]. It has been postulated that TALPID3 regulates the removal of daughter centriole-specific/enriched proteins (DCPs) and promote basal body docking and ciliary vesicle formation[36,42,44–47].

In a whole-genome genetic screen of C. elegans mutants with disrupted ciliogenesis, we retrieved and cloned talp-3, a homolog of mammalian TALPID3. In C. elegans, TALP-3 colocalizes with DYF-19 (a homolog of the TF protein FBF1) and ANKR-26 (a C-terminal homolog of the TF protein ANKRD26) at the basal body. Although single talp-3 or ankr-26 mutants have subtle or no defects in cilia formation and gating, talp-3; ankr-26 double mutants show severely disrupted ciliogenesis and cilia gating. Remarkably, co-depletion of TALP-3 and ANKR-26 completely abolishes the recruitment of DYF-19 to TFs. We further discovered that TALP-3, ANKR-26, and DYF-19 associate in vitro and in vivo. Depletion of TALP-3 or ANKR-26 alone could compromise the in vivo association of the remaining two proteins. Furthermore, we show that human TALPID3 and ANKRD26 share conserved functions with their worm counterparts in orchestrating FBF1 recruitment, ciliogenesis, and cilia gating. Collectively, our findings demonstrate that a highly conserved functional module containing TALPID3-ANKRD26-FBF1 is essential for the proper formation of a functional cilia gate.

## Results

**TALP-3 is a TF-associated protein involved in ciliogenesis.** Among all identified TF components so far, only FBF1 and ANKRD26 are evolutionarily conserved between C. elegans and humans[25,48]. DYF-19 is the C. elegans homolog of FBF1, which plays an essential role in cilia gating[25]. ANKR-26 is encoded by k10g6.4 and is homologous to human ANKRD26 C-terminus, which alone is sufficient for TF localization of human ANKRD26 (Supplementary Fig. 1a–d). More conserved TF or TF-associated components may await identification. We previously performed a genome-wide ethyl methanesulfonate (EMS) mutagenesis screen in C. elegans to search for mutant nematodes with ciliogenesis defects[49]. In C. elegans, mutants with abnormal ciliogenesis cannot take up fluorescent dye and are thus called dye-filling defective (Dyf)[50]. We retrieved hundreds of Dyf alleles, and have been actively mapping the causal loci. One allele, jhu511, was mapped to y57gllc.32 (Fig. 1a; Supplementary Fig. 2a). Protein blast homolog searches against the mouse database revealed that the Y57G11C.32 protein contains a region homologous to the highly conserved region of mouse TALPID3[38] (Supplementary Fig. 2b, c). Based on sequence similarity (Supplementary Fig. 2b–d), subcellular localization, and functional data (see below), we believe that the Y57G11C.32 protein is homologous to mammalian TALPID3; hereafter, we refer to y57g11c.32 as talp-3. jhu511 is a G-A point mutation that alters the splicing donor site of the 2nd intron of the talp-3 gene and creates a putative null allele that encodes a truncated TALP-3 protein with the majority of its amino acid sequence deleted (Supplementary Fig. 2a, e). Surprisingly, contrary to the assumed importance of TALPID3 in mammalian ciliogenesis[46], talp-3 (jhu511) mutants showed only mild ciliogenesis defects with ~20% amphid cilia and ~40% phasmid cilia shortened (Fig. 1b, c) and a subtle reduction in the ciliary IFT machinery (Fig. 1d), suggesting functional redundancy for TALP-3 in the context of cilia. Introduction of the wild-type talp-3 gene rescued the ciliogenesis defect of the jhu511 allele (Fig. 1c). talp-3 (tm7883), an independent allele obtained from the Japanese National BioResource Project (NBRP), encodes a truncated TALP-3, showing similarly mild ciliogenesis defects (Supplementary Fig. 2a, f–h).

Mammalian TALPID3 localizes to the distal end of centrioles[47]. Mutations in TALPID3 lead to defective ciliogenesis in vertebrates[36–38] and cause human JBTS[39–43]. Promoter expression analysis demonstrated that talp-3 is exclusively expressed in ciliated cells in C. elegans (Supplementary Fig. 3a). Consistent with the localization of TALPID3 on the distal end of the centriole in mammalian cells, TALP-3::GFP was found immediately below the TZ marker MKS-5, colocalized with the TF markers DYF-19 and ANKR-26, and partially overlapped with the TF-adjacent protein GASR-8 (a putative homolog of human GAS8) (Fig. 1e; Supplementary Fig. 3b). Interestingly, the highly conserved region in TALP-3 is required for its localization at basal bodies and for its function in ciliogenesis (Supplementary Fig. 3c, d). These results indicate that TALP-3 is located and functions specifically on TFs in C. elegans.

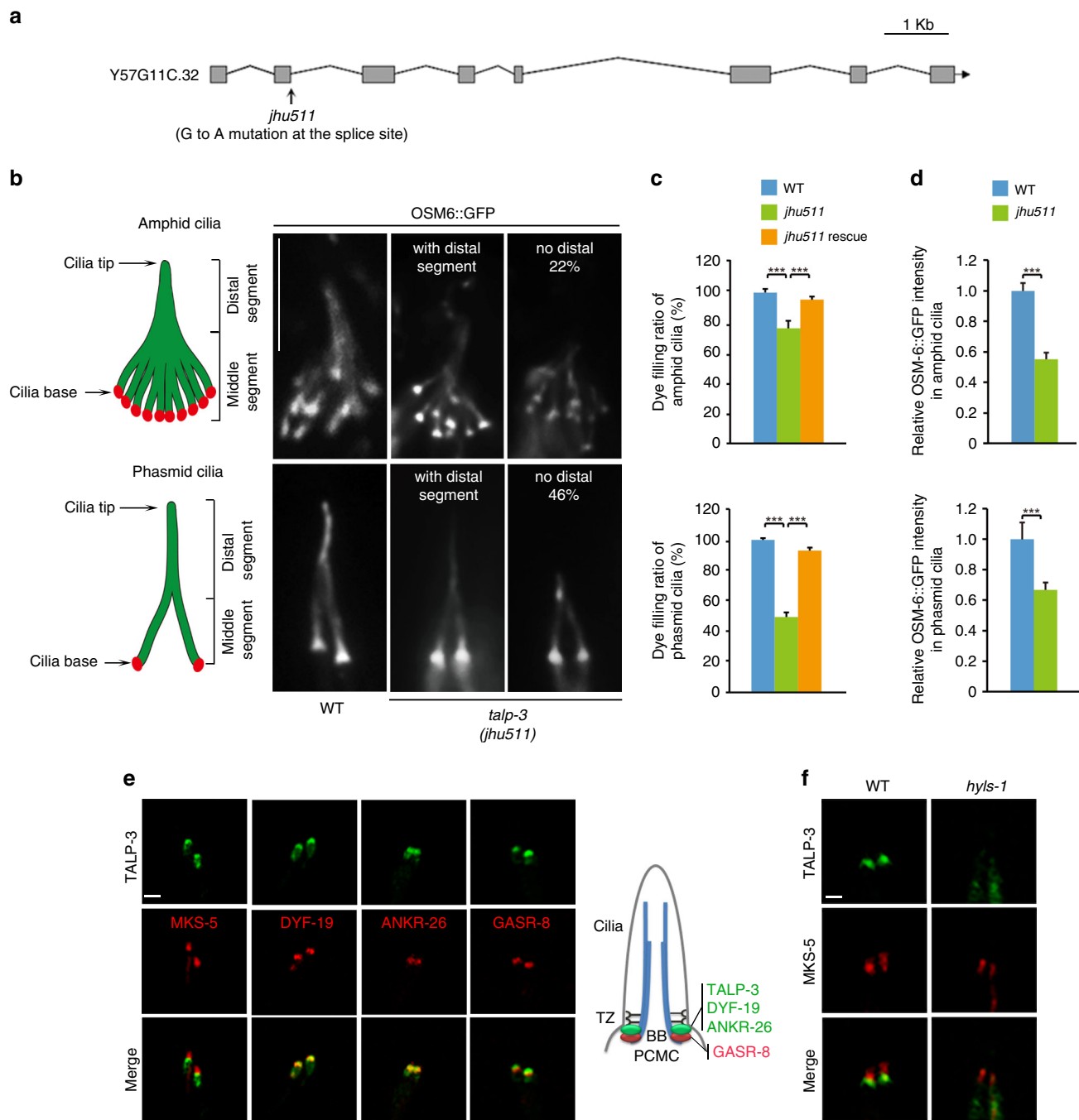

**Fig. 1 TALP-3 is required for ciliogenesis in *C. elegans*. a** Schematic representation of the *C. elegans* Y57G11C.32 genomic structure. *jhu511* mutants obtained from a genome-wide mutagenesis screen for cilia mutants possess a G-to-A mutation at the 2nd intron donor site in the *y57g11c.32* gene. This mutation changes the splice site, resulting in a 155 bp deletion in the 2nd exon, and a reading frameshift. **b** TALP-3 is required for cilia formation. Diagrams showing the amphid cilia and phasmid cilia in *C. elegans* (left panel). Representative images of cilia in WT and *talp-3 (jhu511)* mutants labeled with OSM-6::GFP (right panel). Approximately 22% of amphid cilia and 46% of phasmid cilia in the *jhu511* mutants are truncated. **c** Quantification of the dye-filling ratios of amphid cilia and phasmid cilia in WT, *jhu511* mutant and *jhu511* rescue animals. Ciliogenesis defects in *jhu511* are rescued by introducing a copy of the WT Y57G11C.32 gene. *n* = 300 worms over three independent experiments. **d** Relative fluorescence intensity of OSM-6::GFP in amphid cilia and phasmid cilia in WT and *jhu511* mutant worms. WT, *n* = 39; *jhu511*, *n* = 39. **e** Co-labeling of TALP-3::GFP with the indicated TZ- and basal body-associated proteins tagged with mCherry. TALP-3 localizes immediately below the TZ protein MKS-5 and colocalizes with the TF proteins DYF-19 and ANKR-26. **f** HYLS-1 is required for the proper basal body localization of TALP-3. Data are represented as the mean value ± s.d. Significant differences were identified by two-tailed unpaired Student's *t* test. ***\*\*\*P* < 0.001. Bars, 5 μm in **b**, 1 μm in **e**, **f**. Source data are provided as a Source Data file.

**HYLS-1 regulates the localization of TALP-3 to TFs.** We previously showed that HYLS-1, the ortholog of hydrolethalus syndrome protein 1, is required for TF integrity in *C. elegans*[48]. The identified worm TF components DYF-19 and ANKR-26 and the TF-adjacent protein GASR-8 lost their ability to target the cilia base in *hyls-1* mutants[48]. Accordingly, GFP-tagged TALP-3 lost its specific enrichment in TFs in *hyls-1* mutants, and was dispersed at the periciliary membrane compartment (PCMC), a

subcellular compartment below TFs[51] (Fig. 1f). In contrast, loss of GASR-8, ANKR-26, or DYF-19 did not perturb the localization of TALP-3 (Supplementary Fig. 3c). Notably, TALP-3 deficiency did not affect the localization of ANKR-26 and DYF-19 (Supplementary Fig. 3d), suggesting that although TALP-3 is an exclusive TF-associated protein, it is dispensable for the structural integrity of TFs.

**TALP-3 and ANKR-26 orchestrate cilia gating for IFT proteins**. Considering the mild ciliogenesis defects observed in mutants of both *talp-3* alleles, we hypothesized that another TF protein complements the role of TALP-3 in ciliogenesis. We thus explored the genetic interaction between *talp-3* and other TF genes *dyf-19* and *ankr-26*. Intriguingly, *talp-3; ankr-26* double mutants, but not *talp-3; dyf-19* double mutants, showed synergistic defects in ciliogenesis (Fig. 2a, b; Supplementary Fig. 4a–c). The *ankr-26* single mutant showed completely normal ciliogenesis, and the *talp-3* single mutant had slight defective ciliogenesis; however, all cilia in *talp-3; ankr-26* double mutants were severely truncated, as demonstrated by both the IFT marker OSM-6::GFP and the axonemal tubulin marker TBB-4::mCherry (Fig. 2a, b). Consistent with these results, transmission electron microscopy (TEM) analysis further confirmed that the cilia in *talp-3; ankr-26* double mutants lacked distal segments, but contained partial middle segments (Supplementary Fig. 4b, c). These results suggest that TALP-3 genetically interacts with ANKR-26 to orchestrate ciliogenesis.

Interestingly, we observed that the IFT-B component OSM-6 tended to accumulate at the tips of truncated cilia in *talp-3; ankr-26* double-mutant worms (Fig. 2a; Supplementary Fig. 4d). Gating defects in *dyf-19* mutants lead to defects in the ciliary import of most IFT components, except for IFT-B components, which abnormally accumulate at the tip of truncated cilia[25]. Considering the exclusive localization of TALP-3 and ANKR-26 on TFs, we hypothesized that *talp-3* and *ankr-26* genetically interact to regulate the DYF-19 pathway in cilia gating. To address this hypothesis, we introduced a battery of IFT markers into *talp-3* single, *ankr-26* single, and *talp-3; ankr-26* double mutants, and examined their localization. All IFT components examined showed either normal ciliary signal in *ankr-26* single mutants or slightly reduced cilia signal in *talp-3* single mutants (Fig. 2c). Of note, in *talp-3; ankr-26* double mutants, the IFT-B components OSM-5/IFT88 and IFT-20 also tended to accumulate at the tip of residual cilia (Supplementary Fig. 4d), whereas the IFT-A-associated kinesin-II motor KAP-1, the IFT-A component CHE-11/IFT140, the IFT retrograde motor dynein light chain XBX-1, and the BBSome component BBS-7 were restricted below the ciliary base and failed to enter the cilia (Fig. 2c; Supplementary Fig. 4e). These observations indicated that *talp-3; ankr-26* double mutants completely recapitulate the phenotypes of *dyf-19* mutants, supporting the conclusion that TALP-3 and ANKR-26 function on TFs to orchestrate the proper gating of IFT machinery and control ciliogenesis.

**TALP-3 and ANKR-26 orchestrate cilia import of receptors**. We previously showed that HYLS-1 controls TF formation and the ciliary gating of both soluble proteins (IFT particles) and membrane proteins (sensory receptors)[48]. To determine whether TALP-3 and ANKR-26 are required for the ciliary import of membrane proteins, we examined the localization of PKD2 (a worm ortholog of human polycystin-2), the mechanosensory receptor TRP channel OSM-9, and the G protein-coupled receptor ODR-10 in *talp-3; ankr-26* double mutants. In *C. elegans*, PKD2 localizes to the cilia of male-specific sensory neurons[52], OSM-9 targets the cilia of OLQ mechanosensory

neurons[53], and ODR-10 is expressed only in the cilia of chemosensory AWA neurons[54]. Notably, all sensory receptors examined showed comparable localization in the cilia of WT and *ankr-26* single mutants and subtle mislocalization along dendrites in *talp-3* mutant cilia. In drastic contrast, the specific ciliary enrichment of GFP-tagged PKD2, OSM-9, or ODR-10 was disrupted, and all receptors show strong mislocalization below the cilia base in *talp-3; ankr-26* double-mutant worms (Fig. 2d). The abnormal dendritic accumulation of sensory receptors originates from either a disrupted diffusion barrier at the cilia base that leads to lateral leakage from the cilia membrane to the dendritic membrane or the compromised import of cilia-specific sensory receptors. Taken together, these results indicate that the TF components TALP-3 and ANKR-26 orchestrate the proper ciliary gating of membrane receptors.

**The TALP-3-ANKR-26 module recruits DYF-19 to TFs**. Interestingly, *talp-3; ankr-26* double mutant exhibits gating defects similar to those observed in *dyf-19* single mutants[25]. We thus aimed to explore the functional relationship between DYF-19 and TALP-3/ANKR-26. The localization of TALP-3 or ANKR-26 was not disrupted in *dyf-19* mutant cilia (Supplementary Fig. 4f). In contrast, GFP-tagged DYF-19 completely lost its ability to target TFs in *talp-3; ankr-26* double mutants (Fig. 3a, b), but not in *talp-3* or *ankr-26* single mutants. This suggests that TALP-3 and ANKR-26 cooperate to recruit DYF-19 to TFs to constitute a functional cilia gate. Since these three proteins exhibit similar localizations in *C. elegans* cilia, we speculated that they physically associate with each other. Indeed, GST pull-down assays showed that TALP-3, ANKR-26, and DYF-19 directly interact with each other (Fig. 3c–e). Specifically, the C-terminus of TALP-3 directly interacts with ANKR-26 (Fig. 3c), whereas its N-terminus binds DYF-19 (Fig. 3d).

**TALP-3, ANKR-26, and DYF-19 associate on TFs in vivo**. To determine whether TALP-3, ANKR-26, and DYF-19 form a complex in vivo, we employed the bimolecular fluorescence complementation (BiFC) assay, which directly visualizes the interactions of proteins in the same macromolecular complex in their natural environment[55]. As expected, strong fluorescence complementation among TALP-3, ANKR-26, and DYF-19 was observed specifically on TFs in live animals (Fig. 3f). Notably, TALP-3 and ANKR-26 BiFC signals could be observed in the periciliary membrane compartment below TFs (Fig. 3f), suggesting that they likely form a complex before being targeted to TFs. The TALP-3-ANKR-26 association did not require the presence of DYF-19, whereas depletion of TALP-3 or ANKR-26 partially affected the stability of the complex formed by the remaining two components (Fig. 4a–c). Consistent with the finding that HYLS-1 is required for TF formation, all BiFC signals disappeared in *hyls-1* mutants (Supplementary Fig. 5). Collectively, our data suggest that TALP-3 and ANKR-26 first form a complex and then localize to TFs, where they recruit DYF-19 to form the TALP-3-ANKR-26-DYF-19 module, which primes TFs into a functional ciliary gate for both membrane and soluble proteins. Interestingly, although TALP-3 and ANKR-26 do not have similar protein domains or structures, they show functional redundancy in recruiting DYF-19 to TFs.

**A conserved TALPID3-ANKRD26-FBF1 module in mammalian cilia**. Human TALPID3 mutations cause JBTS, and *Talpid3*$^{-/-}$ mice show typical ciliopathy phenotypes[36]. *Ankrd26*$^{-/-}$ mice show defective cilia signaling and develop obesity, a manifestation associated with ciliopathies[56]. Gaining mechanistic insights into the function of mammalian TALPID3

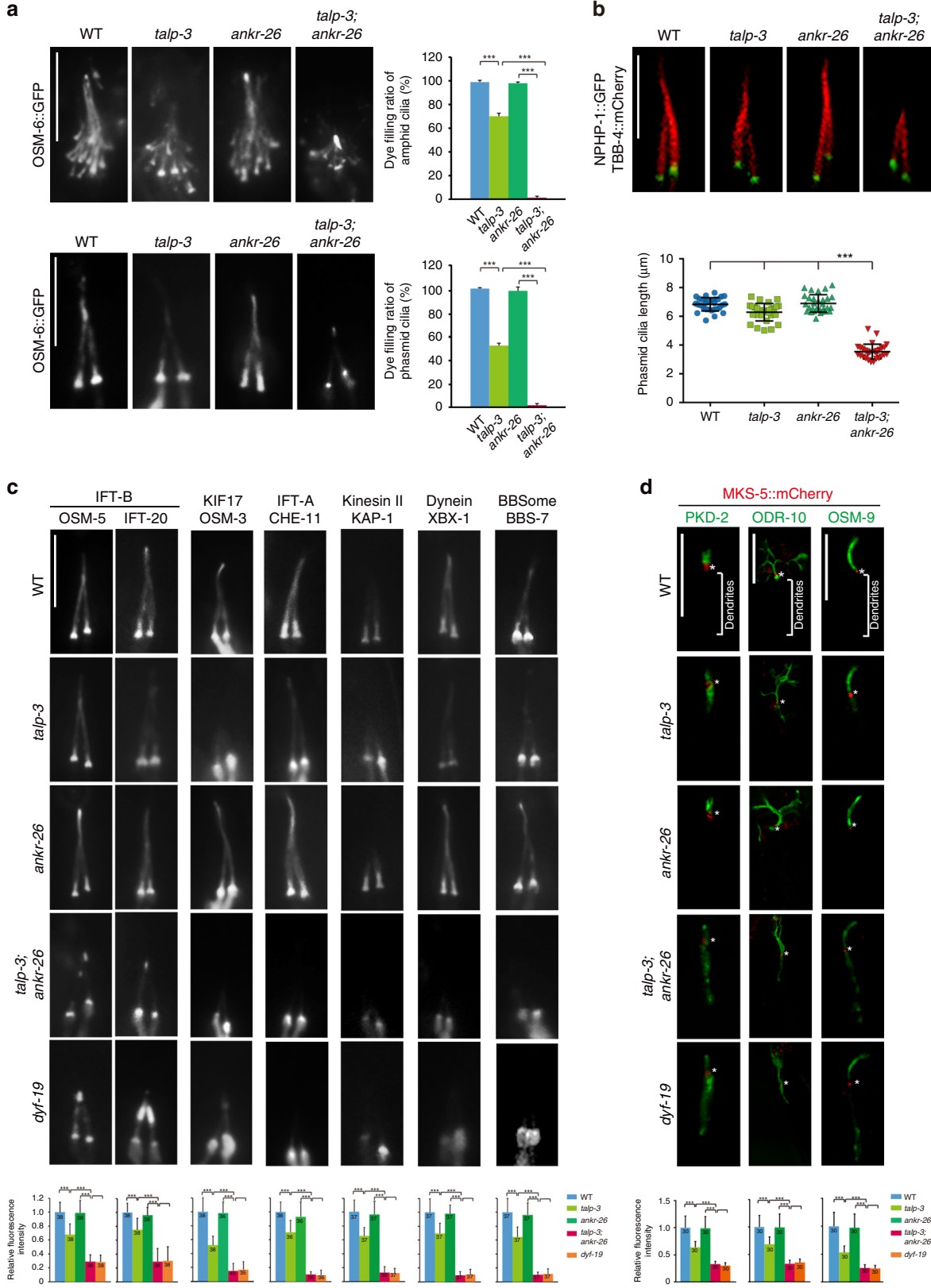

and ANKRD26 would be critical for understanding the pathogenesis underlying corresponding ciliopathies. We first investigated whether TALPID3 and ANKRD26 show any correlation with TFs in mammalian cilia. Both endogenous TALPID3 and overexpressed TALPID3 strongly labeled both centrioles, and ANKRD26 specifically labeled mother centrioles and basal bodies

(Fig. 5a, b). To accurately define their localization on the centriole, we performed super-resolution structured illumination (SIM) microscopy. By co-labeling with the subdistal marker ODF2, we confirmed that TALPID3 and ANKRD26 localize above subdistal appendages and on the same focal plane as TFs (Fig. 5c). TALPID3 localizes to the distal end of both mother and

**Fig. 2 Genetic interaction between TALP-3 and ANKR-26 is required for ciliogenesis. a** Dye-filling assay of amphid cilia and phasmid cilia in WT, *talp-3* single mutant, *ankr-26* single mutant and *talp-3; ankr-26* double-mutant worms. In both amphid and phasmid cilia, *ankr-26* single mutants show no dye-filling defects, and *talp-3* single mutants are partially defective in dye filling, whereas *talp-3; ankr-26* double mutants are completely defective in dye filling. $n = 300$ worms over three independent experiments. **b** Images of phasmid cilia labeled with TBB-4::mCherry and quantification of the corresponding cilia length (lower panel) in worms in the indicated genetic background. The cilia in *talp-3; ankr-26* double mutants are severely truncated. $n = 30$. **c** Localization of various IFT proteins in worms in the indicated genetic background and quantitation of the corresponding relative fluorescence intensity. Compared with that in *talp-3* and *ankr-26* single mutants, the ciliary entry of all IFT proteins in *talp-3; ankr-26* double mutants is severely compromised. Notably, the phenotype of the *talp-3;ankr-26* double mutants is very similar to that of our reported *dyf-19* mutants. The IFT-A component CHE-11, the IFT-A-associated kinesin-II subunit KAP-1, dynein light chain XBX-1, and the BBSome protein BBS-7 fail to enter cilia, whereas some IFT-B components (OSM-5 and IFT-20) can still enter truncated cilia and tend to accumulate at the tips of cilia. Numbers of cilia assayed are indicated in the bars. **d** The ciliary membrane proteins PKD2, ODR-10 and OSM-9 are mislocalized and accumulated in the dendrites of *talp-3; ankr-26* double mutants. Numbers of cilia assayed are indicated in the bars. Stars indicate the TZ. All data are presented as the mean value ± s.d. Significant differences were identified by two-tailed unpaired Student's *t* test. No adjustments were made for multiple comparisons. NS, $P > 0.05$; ***, $P < 0.001$. Scale bars = 5 μm. Source data are provided as a Source Data file.

daughter centrioles, where it forms a "rim" around the centriolar barrel close to the distal appendages[47]. In agreement with this, by using an antibody specifically recognizing the C-terminus of endogenous TALPID3, we observed a "ring" with an ~350 nm diameter that was smaller than the ~450 nm "ring" formed by FBF1. By using FBF1 as a TF marker, we confirmed that both endogenous ANKRD26 and overexpressed Flag-tagged ANKRD26 exclusively labeled TFs (Fig. 5a–d), in agreement with very recent data[57]. SIM microscopy revealed that ANKRD26 forms a ring with a diameter of ~450 nm on TFs similar to the ring formed by FBF1 (Fig. 5d). As expected, ANKRD26, but not TALPID3, lost its localization at the cilia base in CEP83-deficient cells, in which distal appendage formation was disrupted (Fig. 5e). We conclude that ANKRD26 and FBF1 completely colocalize on TFs, whereas TALPID3 localizes more proximal to the centriolar wall (Fig. 5f). Notably, from our SIM analysis, although the mean diameter of TALPID3 rings (~350 nm) was smaller than that of FBF1 or ANKRD26 rings (~450 nm), the rings partially overlapped (Fig. 5d). Interestingly, recent high-resolution characterization of DA organization by stochastic optical reconstruction microscopy revealed that ANKRD26 forms a toroid with an inner diameter ~314 nm and an outer diameter of ~578 nm, but FBF1 forms a toroid with inner and outer diameters of ~269 and ~496 nm, respectively, supporting the notion that TALPID3 partially overlaps FBF1 and ANKRD26[57]. As expected, endogenous coimmunoprecipitation experiments and in vitro GST pull-down assays confirmed that TALPID3, ANKRD26, and FBF1 associate with each other in mammals (Fig. 5g; Supplementary Fig. 6).

**TALPID3 and ANKRD26 coordinate FBF1 centrosomal recruitment**. To assess the function of TALPID3 and ANKRD26 in the context of cilia, we used siRNA to knockdown the *TALPID3* and *ANKRD26* genes in human retinal pigment epithelial (RPE) cells. Knockdown efficiencies were validated by both immunofluorescence staining and immunoblotting (Supplementary Fig. 7a–d). Similar to what was observed in *C. elegans*, depletion of *TALPID3* led to truncated cilia and a reduced ciliation ratio, whereas depletion of *ANKRD26* did not affect ciliogenesis (Fig. 6a). Consistently, co-depletion of *TALPID3* and *ANKRD26* indeed exacerbated ciliogenesis defects, compromised ciliary import of the IFT component IFT140 (Fig. 6a), and significantly reduced FBF1 intensity on TFs (Fig. 6b). We also examined newly synthesized DAs on daughter centrioles. During the cell cycle, the daughter centriole transforms into a new mother centriole and assembles distal and subdistal appendages. FBF1 and other DA proteins are recruited to newly synthesized DAs at this stage. By carefully examining dividing cells, we discovered that simultaneous depletion of *TALPID3* and *ANKRD26*, but not their individual depletion, significantly disrupted FBF1 recruitment to the newly formed mother centriole (Fig. 6c). These

data indicate that TALPID3 and ANKRD26 indeed coordinate the recruitment of FBF1 to the distal appendages of newly formed mother centrioles. Once FBF1 is associated with DAs/TFs, it is likely very stable and does not require TALPID3 and ANKRD26 to maintain its DA/TF localization. Remarkably, none of the other examined DA components (CEP164, CEP89, and SCLT1), proteins localized to the distal part of the mother centriole (ODF2), or TZ components (TCTN1 and CEP290) were affected by co-depletion of *TALPID3* and *ANKRD26* (Fig. 6b, c; Supplementary Fig. 8a, b). These results demonstrate that TALPID3 and ANKRD26 specifically regulate the recruitment of FBF1, but do not affect the overall integrity of DAs/TFs, the TZ, and the distal centriole.

**TALPID3 and ANKRD26 orchestrate cilia entry of receptors**. Various signaling receptors are anchored on the cilia surface, a spatial arrangement that is crucial for the function of cilia as sensory organelles. Although much is known about the critical role of cilia in sensory transduction, little is known about the mechanisms of the proper ciliary import of sensory receptors. For example, the localization mechanism of polycystins mutated in the most common monogenic human disease, autosomal polycystic kidney disease (ADPKD)[58], remains poorly understood. Upon *TALPID3* knockdown, there was a subtle reduction in the percentage of cells with positive PKD2 signal on cilia, while the intensity of PKD2 on cilia decreased about 50%. ANKRD26 deficiency had no effect on either the ratio of PKD2-positive cells or the ciliary intensity of endogenous PKD2. Co-depletion of *TALPID3* and *ANKRD26* dramatically decreased the ciliary entry of PKD2 (Fig. 7a). Similarly, the ciliary localization of endogenous Smoothened (Smo) receptor or the signaling molecule Gli3 was also compromised upon co-depletion of *TALPID3* and *ANKRD26*, but not affected by the depletion of either of the two genes (Fig. 7b, c). Collectively, these data indicate that the essential role of the TALPID3-ANKRD26 module in regulating the cilia gating for sensory receptors and signaling molecules is highly conserved from *C. elegans* to humans.

**Discussion**
TFs anchor basal bodies to the apical membrane and constitute the first visible physical barrier between the cytoplasm and the ciliary lumen[17]. TFs and the adjacent TZ are highly conserved subdomains across ciliated species, and have been proposed as central functional compartments of the proposed ciliary gate[4,18,19]. However, how the ciliary gate forms and controls selective cilia import/export remains unclear. Our previous studies suggested that the TF protein FBF1 is a central player in cilia gating[25]. Here, by using both the *C. elegans* genetic model and mammalian cells, we uncovered that the ciliopathy protein

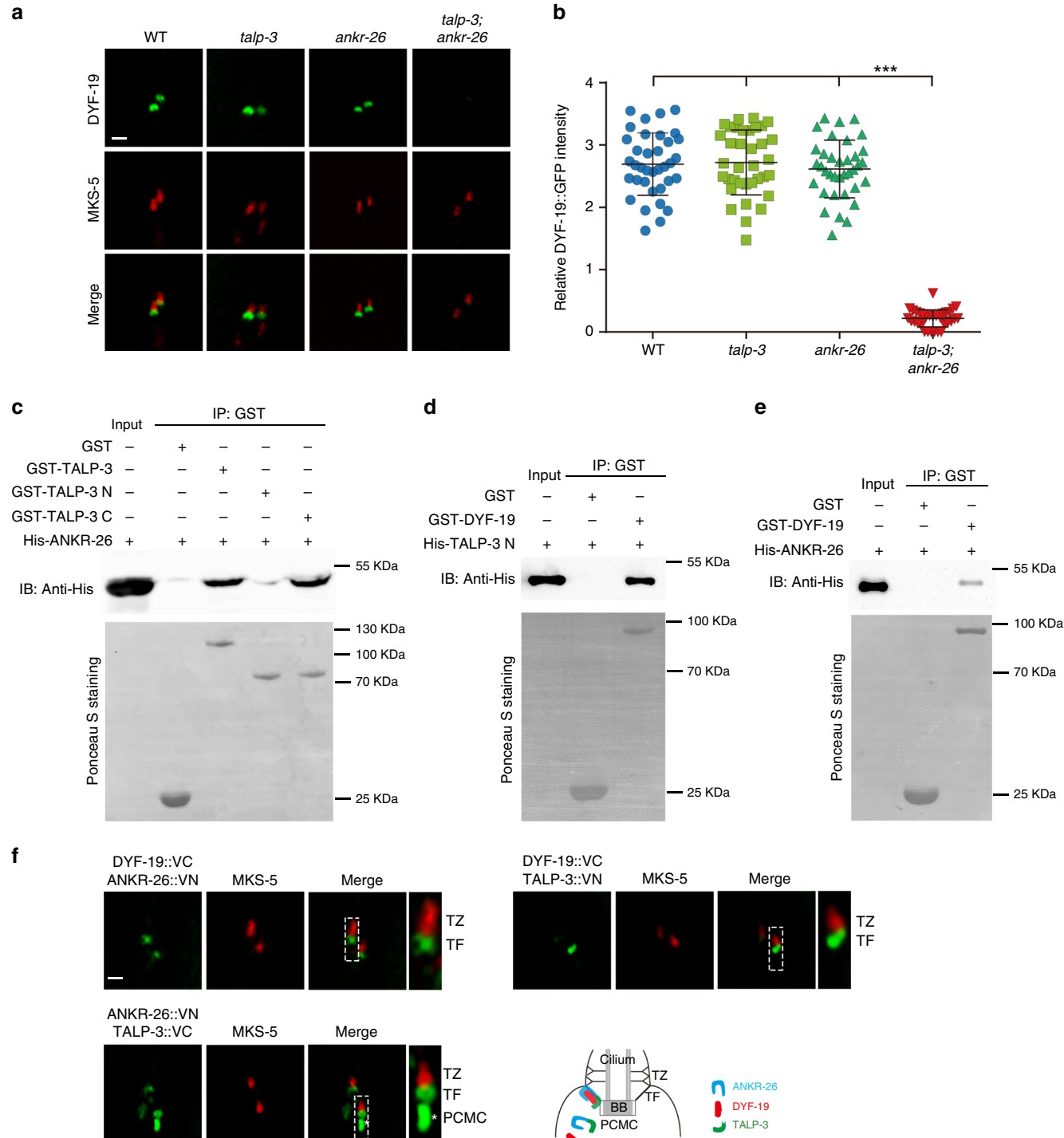

**Fig. 3 TALP-3 and ANKR-26 cooperate to recruit the transition fiber protein DYF-19. a** Localization of the transition fiber protein DYF-19 to the basal body is disrupted in the *talp-3; ankr-26* double mutant. **b** Quantification of relative DYF-19 fluorescence intensity in WT, *talp-3* single mutant, *ankr-26* single-mutant, and *talp-3; ankr-26* double-mutant worms. Data are presented as the mean value ± s.d. *n* = 36. ***, *P* < 0.001 by two-tailed unpaired Student's *t* test. **c** TALP-3 directly interacts with ANKR-26. A GST pulldown assay was used to detect the interaction between His-fused ANKR-26 and GST-fused full-length TALP-3 and truncated TALP-3 constructs (N-terminus, amino acids (aa) 1–450; C-terminus, aa 354 to the C-terminal end). Upper panel, blotted with anti-His antibody; lower panel, loading of the GST and GST-TALP-3 proteins shown by Ponceau S staining. **d** TALP-3 directly interacts with DYF-19 in a GST pull-down assay. **e** ANKR-26 directly interacts with DYF-19 in vitro. **f** TALP-3, ANKR-26 and DYF-19 interact in vivo on TFs. The BiFC assay was performed to visualize the in vivo interaction between TALP-3, ANKR-26, and DYF-19 in living worms. Stable fluorescence complementation between the ANKR-26::VN and TALP-3::VC pair, the ANKR-26::VN and DYF-19::VC pair, and the TALP-3::VN and DYF-19::VC pair was observed on TFs. Notably, strong BiFC signals for ANKR-26::VN and TALP-3::VC are observed in the PCMC region. Scale bars = 1 μm. Source data are provided as a Source Data file.

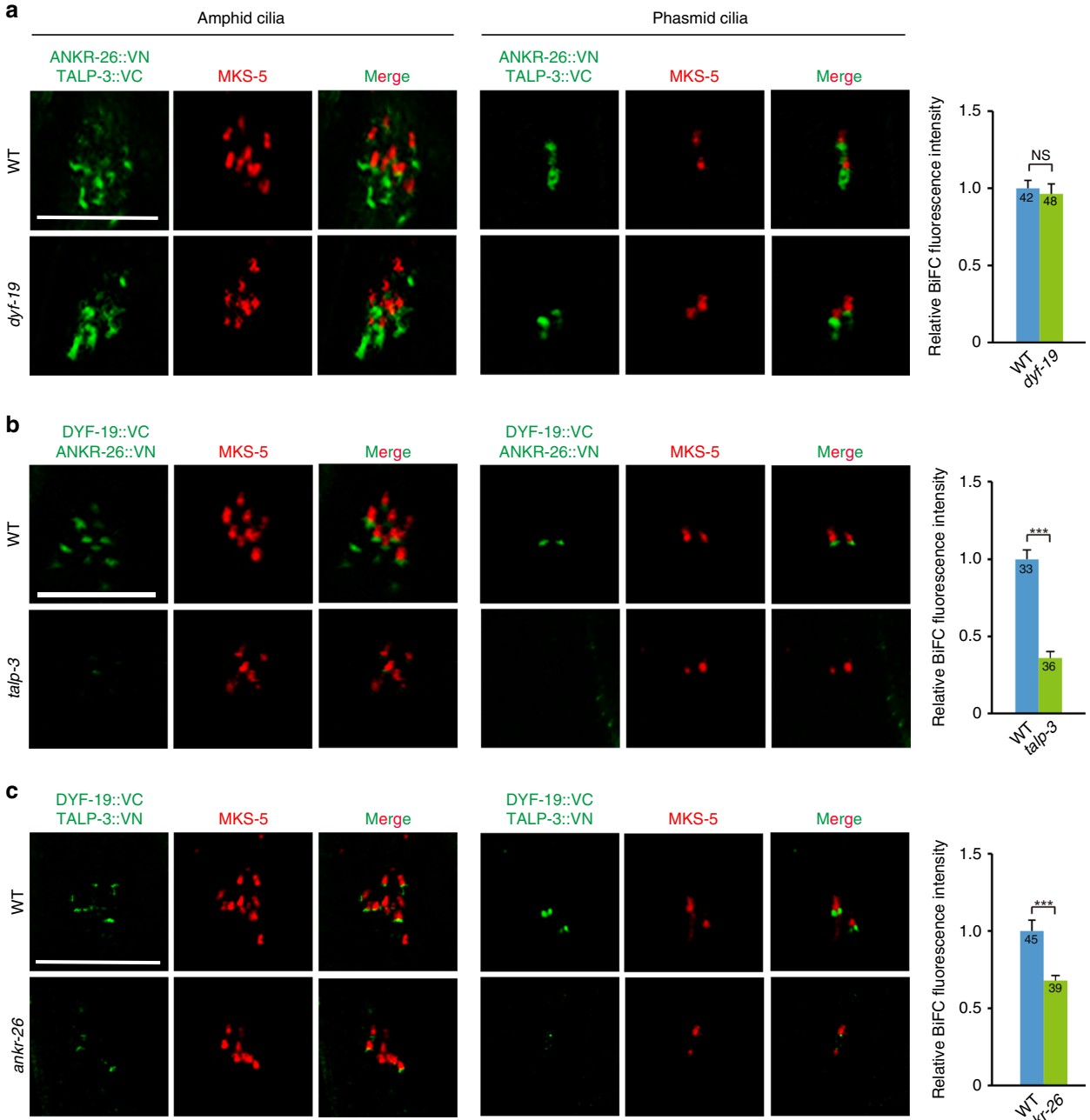

**Fig. 4 In vivo interactions among TALP-3, ANKR-26 and DYF-19 in the corresponding mutants. a** BiFC signal between ANKR-26::VN and TALP-3::VC in amphid cilia and phasmid cilia and relative fluorescence intensity in WT and *dyf-19* mutant worms. Stable fluorescence complementation between ANKR-26::VN and TALP-3::VC is observed in WT worms, and the signal is not impaired in *dyf-19* mutants. **b** BiFC signal between ANKR-26::VN and DYF-19::VC in amphid cilia and phasmid cilia and the relative fluorescence intensity in WT and *talp-3* mutant worms. Stable fluorescence complementation between ANKR-26::VN and DYF-19::VC is observed in WT worms, and the BiFC fluorescence intensity is dramatically decreased in the *talp-3* mutant, suggesting that TALP-3 is required for the proper spatial association between ANKR-26 and DYF-19. **c** BiFC signal between TALP-3::VN and DYF-19::VC in amphid and phasmid cilia, and the relative fluorescence intensity in WT and *ankr-26* mutant worms. Stable fluorescence complementation between TALP-3::VN and DYF-19::VC is observed, and the BiFC fluorescence intensity is decreased in the *ankr-26* mutant, suggesting that ANKR-26 also affects the proper spatial association between TALP-3 and DYF-19 in vivo. All data are presented as the mean ± s.d. Numbers of cilia analyzed are indicated in the bars. NS, $P > 0.05$; \*\*\*, $P < 0.001$ by two-tailed unpaired Student's *t* test. Scale bars = 5 μm. Source data are provided as a Source Data file.

TALPID3 associates with TFs and characterized ANKRD26 as a TF component. In both *C. elegans* and human cilia, TALPID3/TALP-3, ANKRD26/ANKR-26, and FBF1/DYF-19 physically associate with each other to presumably form a TF-specific protein module. The pathway by which TALPID3 coordinates ANKRD26 to promote the TF recruitment of FBF1 and the proper formation of ciliary gate for either soluble or membrane proteins is highly conserved from *C. elegans* to humans. Notably, the TALPID3-ANKRD26-FBF1 protein module specifically localizes on TFs, but not on the TZ, and is essential for the selective import of both membrane and soluble cilia cargoes. Simultaneous depletion of TALPID3 and ANKRD26 or depletion of FBF1 alone, disrupt cilia gating but not affect the overall structure of either TFs or the TZ (Fig. 7d). These data emphasize

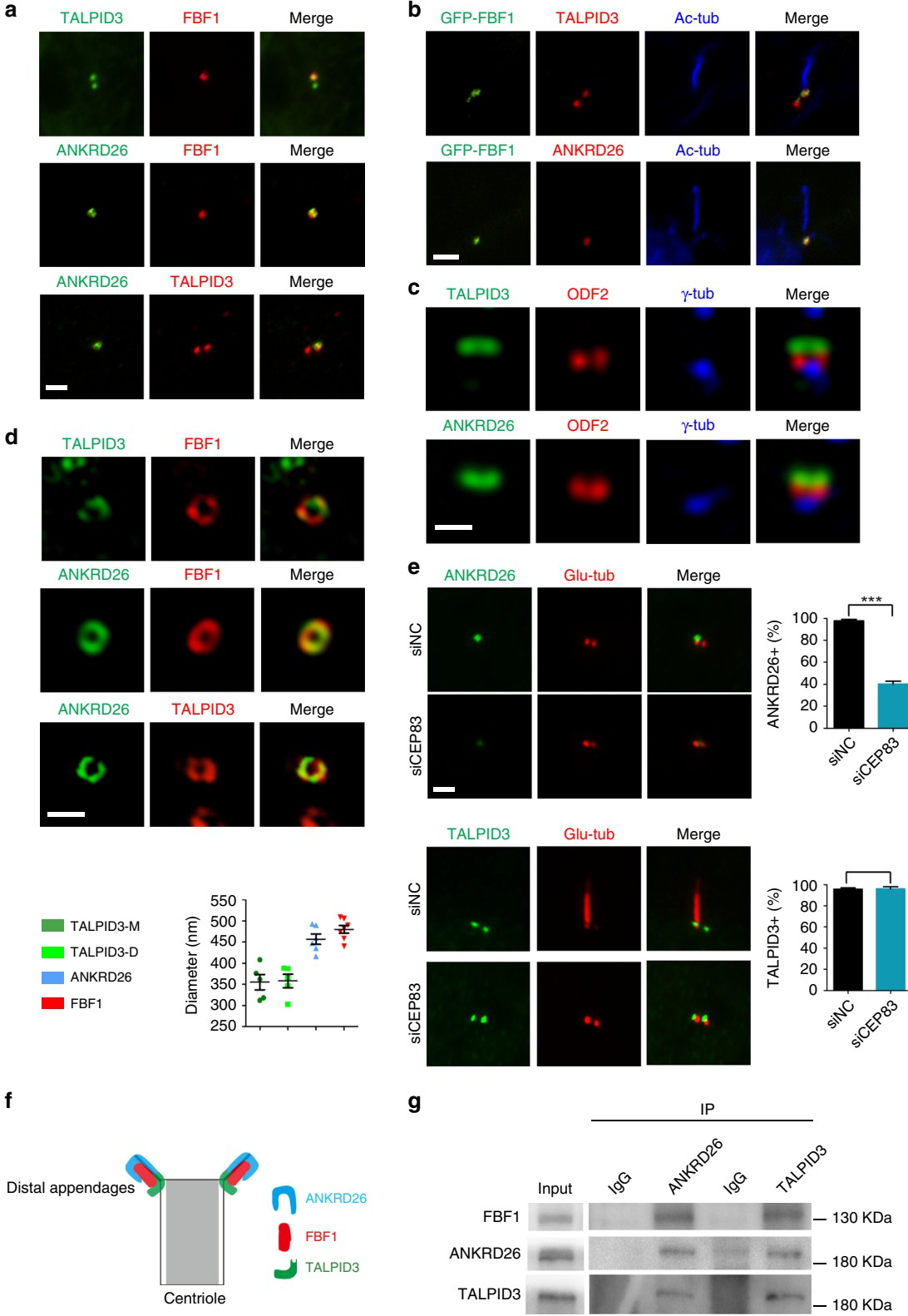

the notion that the TALPID3-ANKRD26-FBF1 module is likely the central functional component of the ciliary gate, and support the idea that TFs, independent of the TZ, could constitute a functional gate for both membrane and soluble proteins[13,18,19].

ANKRD26 was characterized as a centrosome protein required for the cilia import of signaling molecules/receptors in the central nervous system[56]. Our discoveries that ANKRD26 colocalizes with FBF1 and coordinates with TALPID3 to form a functional cilia gate may explain the pathogenesis of mutant phenotypes observed in *Ankrd26*[−/−] mice. Mutations in ANKRD26 have been linked to thrombocytopenia and myeloid malignancies in humans[59–61]. Although primary cilia were believed to be absent from hematopoietic cells based on conjecture, IFT proteins have been implicated as having important roles in immune cells[62,63].

**Fig. 5 TALPID3, ANKRD26, and FBF1 colocalize in the distal centriole in mammalian cells.** TALPID3, ANKRD26 and FBF1 colocalize on the centriole (**a**) and the basal body (**b**). RPE cells were stained with the indicated antibodies, and analyzed by immunofluorescence (IF). **c** Both TALPID3 and ANKRD26 localize above the subdistal appendage protein ODF2. **d** The relative localizations of TALPID3, ANKRD26, and FBF1 on the distal centriole were analyzed with three-dimensional structured illumination microscopy (3D-SIM). The diameters of rings composed of TALPID3, ANKRD26, or FBF1 were measured. TALPID3-M indicates the mother centriole. TALPID3-D indicates the daughter centriole. TALPID3-M, $n = 5$; TALPID3-D, $n = 5$; ANKRD26, $n = 6$; FBF1, $n = 7$. **e** CEP83 is required for the proper localization of ANKRD26 but not TALPID3. Control (siNC) and CEP83-knockdown (siCEP83) RPE cells were stained with the indicated antibodies for IF analysis. $n = 300$ cells over three independent experiments. **f** Cartoons illustrating the putative relative localization of TALPID3, ANKRD26 and FBF1 on DAs. **g** The interactions among FBF1, ANKRD26, and TALPID3 were analyzed using an immunoprecipitation assay. Scale bars, 2 μm in figures showing indirect immunofluorescence, and 0.5 μm in images of 3D-SIM microphages. All data are presented as the mean ± s.d. Significant differences were identified by two-tailed unpaired Student's $t$ test. NS, $P > 0.05$; ***, $P < 0.001$ by Student's $t$ test. Source data are provided as a Source Data file.

In addition, primary cilia have been observed on the surface of human blood and bone marrow cells[64]. With the exclusive localization of ANKRD26 on DAs and TFs, it would be interesting to test whether these diseases are caused by defective cilia gating in the corresponding hematopoietic cells.

TALPID3 is localized to the distal end of both the mother centriole and daughter centriole. Complete loss of *TALPID3* disrupts the removal of daughter centriole-specific/enriched proteins (DCPs), a prerequisite for DA assembly, centriole maturation, and ciliogenesis[46]. However, in agreement with a previous report[46], we noticed that the role of TALPID3 is dosage-sensitive. With ~80% knockdown of TALPID3 in human RPE cells, we observed no or subtle defects in centriole maturation and ciliogenesis, suggesting that residual TALPID3 is likely sufficient to support its postulated functions in centriole maturation, DA formation, and ciliogenesis. From an evolutionary perspective, TALPID3 might not evolve as a central player in centriole maturation since *TALP-3* null worms exhibit completely normal development and proper basal body anchoring and show only subtle defects in ciliogenesis. The conserved coordination between TALPID3 and ANKRD26 in regulating proper formation of the ciliary gate is likely the core function of TALPID3 and a major pathway involved in TALPID3 deficiency-induced ciliopathy phenotypes.

FBF1 is one of the most intriguing TF proteins. In vertebrates, unlike most other TF components, such as CEP164, CEP83, SCLT1, and CEP89, which are essential for basal body docking[32,65], FBF1 is dispensable for ciliogenesis initiation but critical for cilia gating[65]. In invertebrates, among all TF components identified, only FBF1 is conserved in both *C. elegans* and *Drosophila*, and FBF1 is essential for cilia formation in both *C. elegans*[25] and *Drosophila* (our unpublished data). Recently, super-resolution microscopy analyses revealed intriguing feature of FBF1 localization[57,65]. Yang et al.[65] proposed that TFs are composed of fiber blades and matrixes between blades and that FBF1 is the only TF component localized in TF matrixes. While Bowler et al.[57] reported that TFs contain an intricate fibrous base and wider outer sphere head and that FBF1 localizes to the outer head, where it is surrounded by CEP164 and ANKRD26. Although discrepancies regarding the fundamental structure of TFs need to be resolved by more advanced microscopic techniques, both studies suggest FBF1 localizes to a distinct compartment that could be a fundamental part of the functional ciliary gate. Notably, the discovery that the localization of only FBF1, but not other TF components, requires both the centriole wall protein TALPID3 and the TF component ANKRD26 suggests that FBF1-related matrix may exist because it is conceivable that the localization of matrix components likely require coordination between centriole wall and TF blades.

In summary, by using *C. elegans* as a simple but powerful model for studying cilia, we have revealed a conserved role for TALPID3 and ANKRD26 in regulating cilia gating. Although the basal body is degenerated and the conventional fiber-like TF structure may not exist in *C. elegans*[66], the conserved localization of many TF or TF-associated components, including DYF-19/FBF1, HYLS-1, TALP-3, and ANKR-26, suggests that at least, part of TFs or alternative TF homologous structures exist in *C. elegans*. More importantly, the conserved function of these proteins across ciliated species and their tight correlation with ciliopathies suggest that FBF1-related TF is likely the most important subdomain required for cilia gating. Future studies of conserved TF or TF-associated proteins in *C. elegans* have great potential to provide insights into the core function of TFs in the context of cilia and ciliopathies.

## Methods

**C. elegans strains.** All worm strains used in this study are listed in Supplementary Table 1. Standard procedures for the culture and maintenance of *C. elegans* were used[67]. Transgenic animals were generated by microinjection. Standard genetic crossing was used to introduce reporter transgenes from wild-type worms into mutant worms. Polymerase chain reaction (PCR) or sequencing was used to monitor mutant genotypes. Primers used are listed in Supplementary Table 2. *dyf-19 (jhu455)* was used as described before[25]. *hyls-1 (tm3067)*, *gasr-8 (gk1001)*, and *k10g6.4 (gk567)* were obtained from the *C. elegans* Genetic Center (CGC) or the Japanese National Bioresource Project (NBRP). We isolated the *jhu511* mutant during an EMS screen for Dyf mutants as described[49]. The mutation was mapped to chromosome IV using standard single-nucleotide polymorphism mapping techniques. Sequencing of *jhu511* mutants identified a G-to-A mutation at the 2nd intron donor site in *Y57G11C.32*, and ciliary phenotypes of the *jhu511* mutant were rescued upon injection of the *Y57G11.32* CDS. Before phenotypic analyses, the *jhu511* mutant was outcrossed six times to the wild-type (N2) strain.

**Dye-filling assay.** Worms were washed with M9 buffer, and then incubated in the fluorescent lipophilic carbocyanine dye DiI (Sigma-Aldrich, 24364) for 2 h at room temperature. DiI was prepared as a 2 mg/ml stock solution in DMSO and then diluted 1:200 in M9 buffer for use. After incubation with DiI, the worms were washed three times with M9 buffer, and then observed using a fluorescence microscope. Dye filling in amphid was observed with a Nikon SMZ18 stereomicroscope, and dye filling in phasmid was scored under a Nikon Eclipse Ti microscope with a Plan Apochromat ×100 1.49 numerical aperture oil-immersion objective.

**Microscopy and imaging.** Worms were mounted onto 5% agarose pads and anaesthetized using 20 mM levamisole. Images were acquired using either an imaging microscope (Nikon Eclipse Ti or TE 2000-U) with a Plan Apochromat ×100 1.49 oil objective or Olympus FV1000a confocal microscope.

For centrosome staining, cells were fixed with methanol at −20 °C. For cilia staining, cells were fixed with paraformaldehyde at room temperature for 20 min and then permeabilized in 0.2% Triton X-100 for 10 min. After fixation, cells were blocked in 3% BSA and treated with the indicated antibodies. Images were acquired using a Nikon TE 2000-U. Three-dimensional structured illumination microscopy (3D-SIM) was performed following a standard protocol.

**GST pull-down assay.** The pET28a and pGEX-4T-1 vectors were used as backbones to construct plasmids for His- and GST-tagged protein expression, respectively. Primers used are listed in Supplementary Table 2. His- and GST-tagged recombinant proteins were expressed in *Escherichia coli* strain BL21 (DE3), and purified by using Ni-Sepharose beads (GE Healthcare) and GST Sepharose beads (GE Healthcare), respectively. Purified GST or GST fusion protein was immobilized on glutathione Sepharose beads in binding buffer (50 mM Tris-HCl, pH 7.4, 150 mM NaCl, 1% Triton X-100, 1 mM dithiothreitol, 10% glycerol, protease inhibitors),

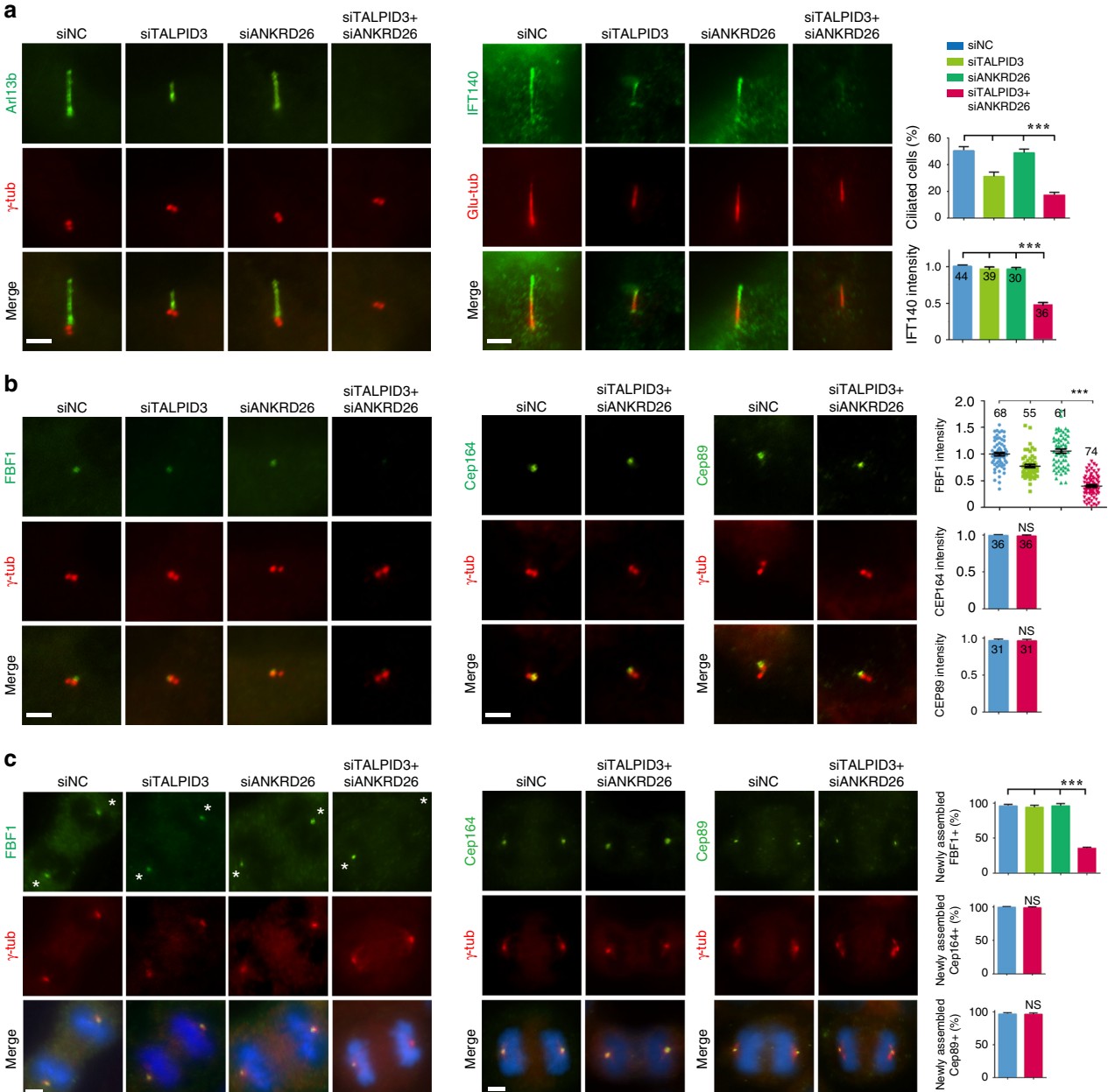

**Fig. 6 The TALPID3-ANKRD26-FBF1 module plays a conserved role in mammalian cells.** RPE cells were treated with negative control siRNA (siNC) or ANKRD26/TALPID3-specific siRNAs (siANKRD26 and siTALPID3, respectively) for 48 h and then subjected to indirect IF with the indicated antibodies after 24 h of serum starvation. **a** Compared with control cells and cells depleted of only TALPID3 or ANKRD26, cells in which ANKRD26 and TALPID3 were both depleted showed significantly enhanced ciliogenesis defects and the compromised ciliary entry of the IFT component IFT140. The percentage of ciliated cells and the relative intensity of IFT140 in the cilia are shown in the right panel. For ciliated cell ratio, $n = 300$ cells over three independent experiments. For IFT140 intensity, numbers of cell examined are indicated in the bars. **b** ANKRD26 and TALPID3 co-depletion impairs FBF1 localization on the distal centriole, but does not affect the localization of CEP164 and CEP89. Quantitative data are shown in the right panel. Numbers of cell examined are indicated in the bars or above each dataset. **c** ANKRD26 and TALPID3 co-depletion disrupts the recruitment of FBF1 to the distal centriole during centriole mutation but does not affect the recruitment of CEP164 and CEP89. Quantitative data are shown in the right panel. $n = 300$ cells over three independent experiments. All data are presented as the mean ± s.e.m. Significant differences were identified by two-tailed unpaired Student's t test. NS, $P > 0.05$; ***, $P < 0.001$. Scale bars, 2 µm. Source data are provided as a Source Data file.

and then incubated with His fusion protein for 4 h at 4 °C. After incubation, the beads were washed with binding buffer five times, loading buffer was added, and the beads were boiled for 10 min. The samples were then subjected to SDS-PAGE and analyzed by western blotting with monoclonal anti-His antibody.

**Bimolecular fluorescence complementation (BiFC) assay.** The Venus-based BiFC assay was used to examine protein interactions in living worms as described[49]. To detect the interactions between TALP-3, ANKR-26 and DYF-19, the following BiFC pairs were used: TALP-3::VC155 and ANKR-26::VN173, DYF-19::VC155 and

TALP-3::VN173, and DYF-19::VC155 and ANKR-26::VN173. The paired BiFC plasmids were co-injected along with the co-injection marker *pRF4 [rol-6 (su1006)]* and the TZ marker MKS-5::mCherry into wild-type animals (a 15 ng/µL concentration for each BiFC plasmid and MKS-5 plasmid and a 100 ng/µL concentration for the pRF4 plasmid). Fluorescent signals were checked using a YFP filter.

**Cell culture and siRNA.** Human retinal pigment epithelial (hTERT RPE-1) and human embryonic kidney (HEK293T) cells were obtained from American Type Culture Collection (ATCC). RPE cells were cultured in the DMEM/F12 medium

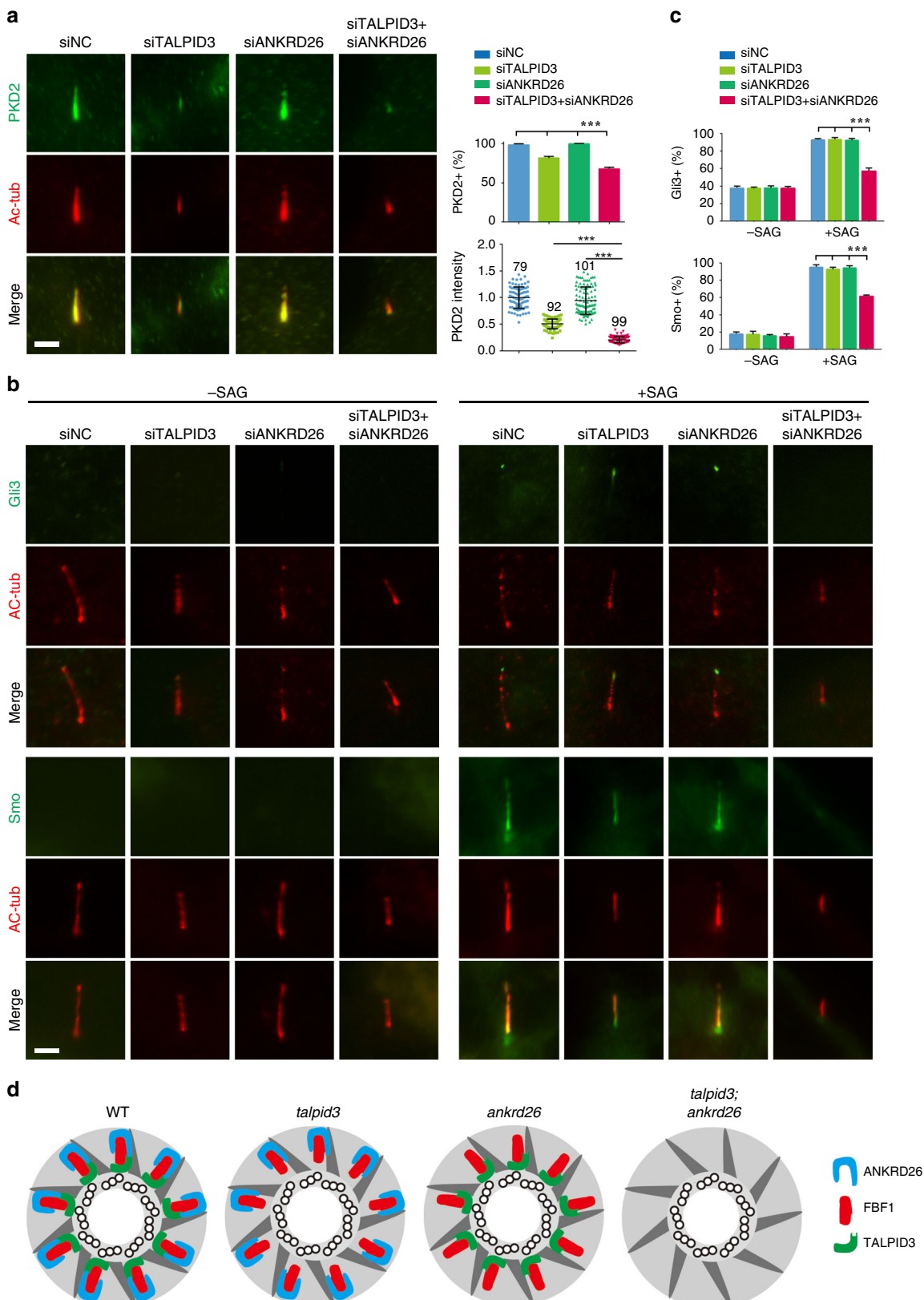

(1:1) supplemented with 10% FBS. HEK293T cells were grown in DMEM containing 10% FBS. To induce cilia formation, RPE cells were starved in DMEM/F12 medium without FBS for 24 h.

All synthetic siRNAs were obtained from Invitrogen and transfected using Lipofectamine RNAiMAX (Invitrogen), according to the manufacturer's instructions. The sequences of the siRNA used were as follows:

siANKRD26-1: GAAUCAAGACUAUGAAUUUtt

siANKRD26-2: CAAGGUUAAUGUACUACAAtt
siTALPID3-1: GUUAAAGGCACUAAGGUAAtt
siTALPID3-2: GGGACUAGUUUGAAUGGAAtt.

**Antibodies**. The following primary antibodies were used: acetylated α-tubulin (T7451, diluted 1:2000 for immunofluorescence, Sigma), γ-tubulin (T6557, Sigma), FLAG (F1804, Sigma), HA (H3663, Sigma), CEP83 (HPA038161, Sigma), CEP164

**Fig. 7 TALPID3 and ANKRD26 orchestrate the cilia gating for membrane receptors.** The localizations of PKD2 and Shh signaling pathway-associated proteins were analyzed in control, TALPID3 single-knockdown, ANKRD26 single-knockdown and TALPID3 and ANKRD26 double-knockdown RPE cells. Cells were examined by IF staining after 24 h of serum starvation with the indicated antibodies. **a** Compared with WT, TALPID3 single-knockdown, and ANKRD26 single-knockdown cells, TALPID3 and ANKRD26 double-knockdown cells showed significantly reduced ciliary entry of PKD2. Right panel, both the percentages of cells with PKD2 ciliary signaling and the mean fluorescence intensity in each group were quantified. For PKD2-positive ratio, $n = 300$ cells over three independent experiments. For PKD2 intensity, numbers of cell examined are indicated above each dataset. **b** Cells treated without (−SAG) or with (+SAG) SAG were compared to analyze Gli3 and Smoothened (Smo) localization. **c** The percentages of cells with ciliary Gli3 and Smo were quantified. Results from a minimum of 100 cells in each group were averaged. The results from three independent experiments were statistically analyzed and plotted. $n = 300$ cells over three independent experiments. **d** A prospective model of the regulation of FBF1 by TALPID3 and ANKRD26. Recruitment of FBF1 to DAs/TFs or its stabilization on DAs/TFs requires coordination between the distal centriole wall and DA blades. Deletion of TALPID3 compromises the condition of the distal centriole wall, and deletion of ANKRD26 may affect the condition of the DA blade; therefore, co-depletion of TALPID3 and ANKRD26 exacerbates defects in FBF1 localization. All data are presented as the mean ± s.d. Significant differences were identified by two-tailed unpaired Student's $t$ test. NS, $P > 0.05$; ***, $P < 0.001$. No adjustments were made for multiple comparisons. Scale bars, 2 μm. Source data are provided as a Source Data file.

(SAB3500022, Sigma), CEP290 (ab84870, Abcam), TALPID3 (24421-1-AP, Proteintech), ARL13b (17711-1-AP, Proteintech), FBF1 (11531-1-AP, Proteintech), TCTN1 (15004-1-AP, Proteintech), SCLT1 (14875-1-AP, Proteintech), IFT140 (17460-1-AP, Proteintech), β-actin (sc-47778, diluted 1:5000 for western blotting, Santa Cruz), SMO (sc-166685, Santa Cruz), CEP89 (ab204410, Abcam), ANKRD26 (GTX128255, GeneTex), polyglutamylated tubulin (ALX-804-885-C100, diluted 1:2000 for immunofluorescence, Enzo Life Sciences), GLI-3 (AF3690, R&D Systems), ODF2 (H00004957-M01, Abnova), and Polycystin-2 (Baltimore Polycystic Kidney Disease (PKD) Research and Clinical Core Center). The secondary antibodies were goat anti-mouse Alexa Fluor 488/594 or goat anti-rabbit Alexa Fluor 488/594 (1:1000 dilution). Primary antibodies were diluted 1:500 for immunofluorescence and 1:2000 for western blotting experiments unless otherwise specified. Uncropped versions of blots can be found in Supplementary Fig. 9.

**Immunoprecipitation.** Immunoprecipitation (IP) was performed using the lysates of HEK293T cells 48 h after transfection in IP buffer (20 mM Hepes-KOH, pH 7.2, 10 mM KCl, 1.5 mM MgCl$_2$, 1 mM EDTA, 1 mM EGTA, 150 mM NaCl, 0.5% NP-40), with Complete Protease Inhibitor Cocktail (Roche) and PhosSTOP Phosphatase Inhibitor Cocktail (Roche) added according to the manufacturer's instructions.

**Transmission electron microscopy.** Young adult worms were fixed in 2.5% glutaraldehyde in cacodylate buffer for 24 h at 4 °C, postfixed in 1% osmium tetroxide in cacodylate buffer, dehydrated in a graded series of ethanol, and embedded in EPON 812 resin, according to standard procedures. Serial sections (~70 nm thickness) were collected from worm heads and examined with an electron microscope (Hitachi H-7650; Hitachi). At least eight worms of each strain were sampled.

**Statistics and reproducibility.** Statistical differences between two samples was determine using two-tailed unpaired Student's $t$ test in Excel. $P$-values > 0.05 were considered not statistically different. $P$-values < 0.001 (marked as ***) were considered statistically significant different. All in vitro experiments were repeated at least three times. All the live image experiments of worms were performed more than 10 times.

**Ethics statement.** Ethical approval is not required in all experiments.

## Data availability
All data supporting the findings of this study are included within the article and its Supplementary Information files or Source Data files. Reagents are available from the corresponding authors upon request.

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

## Acknowledgements

We thank the *Caenorhabditis* Genetics Center, the Japanese National Bioresource Project, and Drs. Maureen Barr, Jonathan Scholey, Bradley Yoder, and Michel Leroux for strains. This work was supported by National Natural Science Foundation of China (Grants 31671549 and 31871357) to Q.W., and National Natural Science Foundation Youth Project of China (Grant 31702019) to H.H.

## Author contributions

Q.W. and J.H. initiate the project and designed experiments. H.Y. and C.C. designed and carried out most of experiments and data analysis. H.C. and H.H. assisted the *C. elegans* experiments. Y.H. assisted the cell experiments. J.H. and Q.W. wrote the paper with the help of Y.H., H.C. and K.L.

## Competing interests

The authors declare no competing interests.
