## [Peer Review File · Nature Communications]

Reviewers' comments:

Reviewer #1 (Remarks to the Author):

This manuscript by the Wei and Hu labs uses two model systems, *C. elegans* and mammalian cells, to shed light into the function of three proteins in building a functional module at the base of cilia. Cilia are evolutionarily conserved organelles that serve two essential functions, namely cell/fluid movement and cellular signaling. The latter function is akin to that of a cellular antenna; the cilia have receptors and signaling machinery within their axonemal compartment, and extracellular signals are transduced into the cell via the cilium. For cilia to form correctly, a basal body (modified centriole) docks onto the cell surface and attaches itself via transition fibers (TFs). The TFs are known to contain a number of different proteins, including FBF1 which is conserved in *C. elegans* and mammals. The TFs are also the location where intraflagellar transport (IFT) machinery can assemble or dock. IFT is a multi-protein transport system required for ferrying cargo into and out of cilia, which is essential for the biogenesis and maintenance of the microtubule-based organelle. TFs may also play a role in preventing vesicles from entering cilia, and thus help to act as a 'gate'. Disruption to this gate, or IFT, or ciliary signaling is associated with a growing number of human disorders, including Joubert Syndrome, collectively termed 'ciliopathies'.

The manuscript shows a very nice complement of techniques to address the nature of the functional interaction between three TF-associated proteins, DYF-19/FBF1, TALP-3/TALPID3 (approved HGNC symbol is not TALPID3, it is KIAA0586) and ANKR-26/ANKRD26. This includes a genetic screen, biochemistry to confirm protein-protein interactions, cell biology (localization in wildtype and mutant cells) and innovative techniques such as bimolecular fluorescence complementation (BiFC) assay to assess *in vivo* protein interactions (in *C. elegans*) and super-resolution microscopy in mammalian cells. The use of two complementary systems (*C. elegans* and mammalian cells) is a very strong aspect of this study. Controls, for example, Fig. S5 showing the efficiency of gene silencing by Western blot analysis, are on the whole included and help support the findings and conclusions of the study. This study also makes an attempt to provide insights into what the molecular and potential physiological effects of disrupting the TF proteins has, largely by observing several ciliary proteins known to be linked to ciliopathies, e.g., PKD2, Smo (Hedgehog signaling).

There are some issues which cloud the findings of the *C. elegans* results (see below), which will need to be addressed. But on the whole, this study has identified a previously unknown trio of proteins that functions at the TFs to influence the function of cilia, findings that are not only relevant to those interested in cilia, but are relevant to human health and disease.

MAJOR ISSUES

1. The manuscript is on the whole poorly written, with numerous grammatical errors. Case-in-point, the first sentence of the introduction is not grammatically correct. Neither is the second sentence. Other errors are found in this first paragraph, and other issues include not defining the meaning of 'primary cilia' (what are the other types of cilia?). Altogether, this makes the manuscript difficult to read. Some of the issues can only be attributed to carelessness, and perhaps no proofreading; for example, the *talp-3* gene is referred to as *taplid-3* on p. 9, which represents a weird cross between *C. elegans* and mammalian nomenclature. I have made no attempt here to correct any of these errors, as there are simply too many; it is up to the authors to ensure that their paper is proof-read and corrected.
2. *C. elegans* Y57G11C.32 is not clearly the homolog/ortholog of TALPID3, as the authors claim. HGNC does not list it as a homolog, and my own homology searches do not identify TALPID3 when the sequence of Y57G11C.32 is used as a query. Instead, this appears to be a worm/nematode-specific protein. The authors need to demonstrate why they believe this *C. elegans* gene encodes a TALPID3

homolog/ortholog.

3. *C. elegans* K10G6.4, which the authors claim is the homolog/ortholog of ANKRD26, is also not mentioned in HGNC as being the mammalian protein homolog, and my own homology searches reveal that this protein also seems to be worm/nematode-specific. The authors state that the proteins are homologous, but do not show this. Neither Y57G11C.32 or K10G6.4 are listed in Wormbase, which is the official curator of *C. elegans* genetic (and other published/unpublished) information.

4. Fig. 2c. It is unclear, from just using IFT markers, whether cilia are still present. Are IFT proteins restricted from entering cilia in the *talp-3;ankr-26* double mutants, or is it that cilia are not present? The authors that TALP-3 and ANKR-26 act to 'gate' IFT proteins, i.e., control their entry into cilia. But if cilia are not present. Fig. 2d suggest that there might be some *types* of cilia still present (partially present?) in the double mutant worms, but these are not the same cilia shown in Fig. 2c. In these images, different receptors are claimed to be mislocalized, but it is unclear where the base of the cilium is in each case; a co-marker would be necessary to establish this and clearly delineate where the cilia begin (if at all).

The authors need to properly show whether it is a wholesale ciliogenesis defect present in the double mutant (*talp-3;ankr-26*), i.e., cilia are absent, or they are still present and IFT does not (perhaps frequently) enter cilia while still permitting the microtubule axoneme to be present. Comparative electron microscopy (TEM) of the *talp-3*, *ankr-26* and *talp-3;ankr-26* strains would be necessary to show whether cilia are present or not, and if they are present, what ultrastructural perturbations they may have.

OTHER ISSUES

1. p. 8, 9: It is not at all clear that OSM-6 accumulates at the tip of cilia based on the images shown, and there is no quantification to back up this claim. Similarly, the authors mention ciliary tip accumulation of OSM-5 and IFT-20 in the *talp-3;ankr-26* mutant. Yet, I see a very similar 'accumulation' of OSM-5 in *ankr-26* cilia, and of IFT-20 in WT cilia. The authors are making claims without proper supporting, quantitated data.

2. How do the authors explain their super-resolution microscopy results, where in mammalian cells there does not appear to be a complete co-localization of ANKR26, FBF1 and TALPID3? If they in fact form a complex, the relatively small sizes of the proteins would likely mean that their fluorescent tags should (almost completely) overlap.

MINOR ISSUES

1. References in the introduction are dated and therefore less useful for the reader. Many review articles are from 2002-2009, and these miss important new developments (e.g., dealing with ciliopathies).

2. p. 12: obesity is not a 'common' phenotype associated with ciliopathies; it is one of several phenotypes (retinal degeneration, cystic kidneys are more common for example).

Reviewer #2 (Remarks to the Author):

This manuscript extends a prior study by Jung-Chi Liao's lab (Nat. Comm 2018) that proposed a role for the transition fiber protein FBF1 in retention of membrane proteins in cilia. This gate function for the TF has received some indirect support by another study (Ye, Nager & Nachury, JCB 2018) that

described a diffusion barrier at the base of cilia with dimensions consistent with the TFs. However, besides these two very recent papers, there has been little evidence for a role of the transition fibers in forming a ciliary gate. The current study represents a significant advance in describing two TF components, TALPID3 and ANKRD26 that function redundantly in retaining ciliary membrane proteins in cilia. The study is of high experimental quality and the major conclusions are doubly buttressed by conducting experiments in both nematodes and mammalian cells. Another strength of the paper is the combination of genetics, biochemistry and imaging used to arrive at the main conclusions of the paper. My concerns are all minor and most can be addressed in writing.

Minor points:

The ciliary levels of the membrane proteins PKD-2, ODR-10 and OSM-9 in Fig. 2d need to be properly measured and plotted.

'FBF1 is the only one conserved from *C. elegans* to human and indispensable for TF assembly'. Do the authors mean to say that FBF1 is dispensable to TF assembly?

Can DYF19 be renamed FBF1? The same question applies to the IFT genes. What purpose does it serve to use the original names when work in every other organism besides nematode uses a universal nomenclature of cilia proteins. This worm exclusivity makes worm studies less accessible to a large audience.

'This suggests that TALP-3 and ANKR-26 cooperate to recruit DYF-19 to TFs'. Typo on cooperate

'We first investigated whether TALPID3 and ANKRD26 show any correlation with TFs'. Is correlation the correct word?

'TFs as a whole, independent of the TZ, could constitute a functional gate for both membrane and soluble proteins.' This idea was previously proposed by Ye, Nager and Nachury in a 2018 JCB paper. It has also been covered in various reviews.

'Completely loss of talpid3 disrupts the removal of DCP, a process indirectly blocks the subsequent DA assembly, centriole maturation, and downstream ciliogenesis'. This is not an easy sentence to read. DCP were defined 13 pages before and never mentioned in between. The sentence appears grammatically incorrect, possibly a word missing?

Fig. 7a: measurement of intensity need to show individual data points as in 3b.

Reviewer #3 (Remarks to the Author):

Wei and Hu labs reported that the ciliopathy protein TALPID3 and ANKRD26 regulate cilia gating. The formation of a functional cilia gate is a fundamental problem in cell biology and ciliopathy. Using both *C. elegans* and mammalian cells, they identified that TALPID3/TALP-3 is a transition fibers-associated component. Double mutant analyses showed that TALP-3 and ANKR-26 function together in cilia gating. Their biochemical data and BiFc experiments nicely showed that TALP-3 and ANKR-26 form a complex with FBF1/DYF-19. All the experiments are well performed and their data are convincing. The study provide an important advance for our understanding of cilia gate and the publication of the manuscript on Nature Communication will be a nice addition the journal.

The following minor issues in the writing need to be addressed;

1) Each sentence in the Introduction needs a reference. The first paragraph on page 5 needs to be

fixed.

2) The term of null should be carefully used. There should be more experiments to support that both alleles are null (e.g. Western or RNA-seq to exclude alternative splicing etc). "putative null" can be safe in many cases without additional experiments.

3) To support "complete colocalization", quantifications of figure 1 (e.g. line-scan of fluorescence) are required.

4) Language can be improved. For example, "When use GFP-tagged IFT-B component OSM-6 as a marker".

5) C.elegans should be C. elegans

Reviewer #1:

MAJOR ISSUES:

Question 1. The manuscript is on the whole poorly written, with numerous grammatical errors. Case-in-point, the first sentence of the introduction is not grammatically correct. Neither is the second sentence. Other errors are found in this first paragraph, and other issues include not defining the meaning of 'primary cilia' (what are the other types of cilia?). Altogether, this makes the manuscript difficult to read. Some of the issues can only be attributed to carelessness, and perhaps no proofreading; for example, the *talp-3* gene is referred to as *taplid-3* on p. 9, which represents a weird cross between *C. elegans* and mammalian nomenclature. I have made no attempt here to correct any of these errors, as there are simply too many; it is up to the authors to ensure that their paper is proof-read and corrected.

We sincerely apologize for our writing. We revised the manuscript and had it proof-read by a commercial language-editing service.

Question 2: *C. elegans* Y57G11C.32 is not clearly the homolog/ortholog of TALPID3, as the authors claim. HGNC does not list it as a homolog, and my own homology searches do not identify TALPID3 when the sequence of Y57G11C.32 is used as a query. Instead, this appears to be a worm/nematode-specific protein. The authors need to demonstrate why they believe this *C. elegans* gene encodes a TALPID3 homolog/ortholog.

We appreciate the concern raised by the reviewer and address it here by emphasizing the following evidence:

1. Y57G11C.32 contains the highly conserved region of TALPID3.

When we mapped Y57G11C.32, we used the BLAST tool to search for the homologous protein, and mouse TALPID3 was identified as one of the top hits (NCBI PSI-BLAST; Organism: *Mus musculus* (taxid:10090)). Also, Y57G11C.32 was identified as the nematode homolog of mouse TALPID3 in a reverse search.

We agree with the reviewer that the similarity between Y57G11C.32 and TALPID3 (29.02% when blast the full length of Y57G11C.32) is indeed low. This is actually very common for the homologs between nematode and mammalian proteins. Even for the genes involved in highly conserved ciliary programs, sequence similarity tends to be poor due to hundreds of million years of evolution from the nematode to mammals (Please see Table 1 below). The similarity between Y57G11C.32 and TALPID3 is actually comparable to that of other *conserved centriolar/ciliary proteins* (as shown in Table 1 below). In many cases, worm scientists need to integrate data from the analyses of subcellular localization, functional analyses, and protein interactome to confirm the homology.

Nonetheless, the most important protein domains are usually kept during evolution regardless of poor homolog similarity across ciliated species. For example, nematode LOV-1 and its

mammalian ortholog Polycystin 1 only share 24.68% similarity along its 3,284 aa protein sequence, but the PLAT domain (about 100 amino acid) is highly conserved; nematode SAS-6 and mammalian SAS-6 only share 29.47% similarity, but the N-terminal PISA motif is highly conserved.

To date, only one highly conserved region (aa 485–568) of mouse TALPID3 has been identified (ref. 39, Yin et al. *Development* 136, 655-664 (2009)) and shown to be essential for centrosome localization of vertebrate TALPID3. It is proposed that this domain, although with unclear function, is critical for the role of TALPID3 in the context of cilia. Of note, BLAST found that the greatest similarity in Y57G11C.32 and TALPID3 lies at aa 425–538 of Y57G11C.32 and aa 459–568 of mouse TALPID3 containing the highly conserved region (Supplementary Fig. 2b, c), and Y57G11C.32 is the sole nematode protein contains this domain, suggesting Y57G11C.32 is probably the only nematode homolog of TALPID3 during evolution. Furthermore, we demonstrated that this region is also important for the localization and function of Y57G11C.32 (Supplementary Fig. 3c, d).

We notice mammalian TALPID3 contains longer C-terminus, which likely endows TALPID3 new function during evolution. However, it was reported that the N-terminus of TALPID3 is required for centriole maturation and DA assembly as well as IFT recruitment, whereas its C-terminal half likely regulates vesicle docking and ciliogenesis initiation (ref. 47, Wang et al., *Nature communications* 9, 3938 (2018)). Intriguingly, in agreement with the fact that the nematode Y57G11C.32 is shorter and homologous to the N-terminal half of TALPID3, Y57G11C.32 is required for cilia integrity but dispensable for ciliogenesis initiation in *C. elegans*.

Table 1. Similarity between *C. elegans* and human centriolar/ciliary proteins.

	C. elegans	Human	BLOSUM62	
			% Identity	% Similarity
TF or TF associated proteins	TALP-3	TALPID3 (aa 1-850)	13.11	29.02
	ANKR-26	ANKRD26 (aa 779-1227)	17.52	39.32
	ANKR-26	ANKRD26 (aa 1222-1710)	19.04	38.65
	DYF-19	FBF1 (aa 500-1133)	12.69	33.28
Centriole proteins	HYLS-1	HYLS1	16.56	31.12
	SAS-6	SAS-6	12.78	29.47
	SAS-4	SAS-4/CPAP	9.93	21.73
	SPD-2	SPD-2/CEP192	5.51	12.41
Basal body proteins	K04F10.2	KIAA0556	11.24	17.68
	GRDN-1	Girdin	16.12	32.46

	CHE-10	Rootletin	16.92	37.24
TZ proteins	CCEP-290	CEP290	14.19	31.08
	NPHP-2	INVS/NPHP2	15.44	29.77
	NPHP-4	NPHP4	20.69	37.30
Membrane receptor	LOV-1/PKD-1	PKD1	11.89	24.68
IFT component	IFT-43	IFT43	15.31	32.88

2. The subcellular localization and function of Y57G11C.32 are similar to those of TALPID3.

Mammalian TALPID3 localizes to the centriole distal end close to the distal appendages/transition fibers and is required for ciliogenesis. Consistently, in *C. elegans*, Y57G11C.32 localizes to the transition fiber region and is required for cilia formation.

Collectively, judging by the sequence similarity, a highly conserved domain, and conserved subcellular localization and function, we conclude that Y57G11C.32 is the homolog of TALPID3.

Question 3. *C. elegans* K10G6.4, which the authors claim is the homolog/ortholog of ANKRD26, is also not mentioned in HGNC as being the mammalian protein homolog, and my own homology searches reveal that this protein also seems to be worm/nematode-specific. The authors state that the proteins are homologous, but do not show this. Neither Y57G11C.32 or K10G6.4 are listed in Wormbase, which is the official curator of *C. elegans* genetic (and other published/unpublished) information.

In WormBase WS220 version

(<http://ws220.wormbase.org/db/gene/gene?name=WBGene00019641;class=Gene>),

human ANKRD26 was suggested to be the best BLASTP matches of K10G6.4 (see webpage screenshots below).

Mammalian ANKRD26 has four Ankyrin repeats on its N-terminus and a large coiled-coil domain on its C-terminus. Both information from the WormBase WS220 version and our own sequence alignment suggest that K10G6.4 is homologous to both the first half and the second half of the coiled coil domain of ANKRD26 (Supplementary Fig. 1a-c). The similarity between K10G6.4 and the first half coiled-coil (aa 779–1227) of human ANKRD26 is 39.32%, and the similarity between K10G6.4 and the second half coiled-coil (aa 1222–1710) of human ANKRD26 are 38.65%. The conservations are similar to other *C. elegans* centriolar/ciliary gene homologs as discussed in previous concern (see Table 1 above). It appears that ANKRD26 likely first evolved as a pure coiled-coil protein localizes specifically on TFs at the ciliary base. During evolution, it gains additional Ankyrin repeats and duplicates its coiled-coil domain by either gene

fusion or translocation.

Gene Summary for K10G6.4

<http://ws220.wormbase.org/db/gene/gene?name=WBGene00019641;class=Gene>

Species	Hit	Description	BLAST E-value	% Length
C. brenneri	CN:CN32706	gene CBN31820	6.61e-173	99.8%
C. briggsae	BP:CBP10870	CBG18445	8.99e-163	99.8%
C. remanei	RP:RP25163	gene CRE10405	3.8e-115	69.9%
C. japonica	JA:JA41351	gene CJA18529a	9.4e-113	81.9%
P. pacificus	PP:PP31367	gene PPA18555	2.2e-39	97.4%
T. vaginalis	TR:A2ENS5	Putative uncharacterized protein	1.6e-25	91.5%
S. cerevisiae	SGD:YDL058W	Essential protein involved in the vesicle-mediated ER to Golgi transport step of secretion; binds membranes and functions during vesicle docking to the Golgi; required for assembly of the ER-to-Golgi SNARE complex	2.2e-23	94.8%
O. cuniculus	SW:P37709	Trichohyalin	3.2e-20	98.9%
C. elegans	WP:CE43383	DYF-14, isoform d	9.4e-20	98.3%
D. melanogaster	FLYBASE:CG5020	Flybase gene name is CLIP-190-PA;	2e-18	98.7%
H. sapiens	ENSEMBL:ENSP00000365238	Isoform 2 of Ankyrin repeat domain-containing protein 26	6.1e-18	98.7%

Note: BLASTP matches are against the longest protein product.
[view full list] [view alignments]

[BLASTP table] [BLASTP diagram]

Hit	Species	Description	E Value	Source Range	Target Range
ENSEMBL:ENSP00000365238	H. sapiens	Isoform 2 of Ankyrin repeat domain-containing protein 26	6.1e-18	5..458	779..1227
ENSEMBL:ENSP00000357794	H. sapiens	Trichohyalin	6.4e-18	10..455	214..664
HI:HIT000000791	H. sapiens	Isoform 1 of Ankyrin repeat domain-containing protein 26	9.1e-18	5..458	1221..1669
ENSEMBL:ENSP00000365255	H. sapiens	Isoform 3 of Ankyrin repeat domain-containing protein 26	9.1e-18	5..458	1222..1670

We reasoned that, although categorized as an Ankyrin-repeat containing protein, the coiled-coil domain is critical for the localization and function of ANKRD26. Thus, we generated several C-terminal truncations of human ANKRD26 and investigated their subcellular localization and function. Interestingly, we showed that the coiled-coil domain is sufficient for its centrosome localization (Supplementary Fig. 1d). Notably, the second half coiled-coil domain alone could be targeted to the centrosome.

Although the whole sequence similarity is low, based on similarity in coiled-coiled domain, similar subcellular localization (localizes to TF) and similar function (involved in FBF1 localization), we suggest that K10G6.4 is homologous to the C-terminus of ANKRD26, which is

likely a critical fragment of ANKRD26 that mediates its TF-related and FBF1-related function.

Question 4. Fig. 2c. It is unclear, from just using IFT markers, whether cilia are still present. Are IFT proteins restricted from entering cilia in the *talp-3;ankr-26* double mutants, or is it that cilia are not present? The authors that TALP-3 and ANKR-26 act to 'gate' IFT proteins, i.e., control their entry into cilia. But if cilia are not present. Fig. 2d suggest that there might be some *types* of cilia still present (partially present?) in the double mutant worms, but these are not the same cilia shown in Fig. 2c. The authors need to properly show whether it is a wholesale ciliogenesis defect present in the double mutant (*talp-3;ankr-26*), i.e., cilia are absent, or they are still present and IFT does not (perhaps frequently) enter cilia while still permitting the microtubule axoneme to be present. Comparative electron microscopy (TEM) of the *talp-3*, *ankr-26* and *talp-3;ankr-26* strains would be necessary to show whether cilia are present or not, and if they are present, what ultrastructural perturbations they may have.

To answer this question, we performed new co-labeling experiments with the axoneme marker TBB-4::mCherry and TZ marker NPHP-1::GFP, and measured the length of the TBB-4 signal. As shown in Fig. 2b, in *talp-3;ankr-26* double mutants, cilia are present but are significantly shorter than WT, indicating that truncated cilia do exist in double mutants.

We further confirmed this conclusion by TEM as reviewer suggested (Supplementary Fig. 4b, c). In *talp-3;ankr-26* double mutants, no axonemes were observed in the distal pore of amphid channels, but were frequently observed in the middle pore, indicating that partially truncated cilia are present in the double mutants.

In these images, different receptors are claimed to be mislocalized, but it is unclear where the base of the cilium is in each case; a co-marker would be necessary to establish this and clearly delineate where the cilia begin (if at all).

We thank the suggestion by the reviewer. We re-did the experiment by generating new transgenic WT and mutant animals expressing the TZ marker MKS-5::mCherry. Analyses of the new transgenic animals support our major conclusion. The new data are included in Fig. 2d in the revised version.

OTHER ISSUES

1. p. 8, 9: It is not at all clear that OSM-6 accumulates at the tip of cilia based on the images shown, and there is no quantification to back up this claim. Similarly, the authors mention ciliary tip accumulation of OSM-5 and IFT-20 in the *talp-3;ankr-26* mutant. Yet, I see a very similar 'accumulation' of OSM-5 in *ankr-26* cilia, and of IFT-20 in WT cilia. The authors are making claims without proper supporting, quantitated data.

We agree with the reviewer that quantification is the best way to support the conclusion. In revised version, we calculated the distal/middle signal intensity ratio of OSM-6, OSM-5 and IFT-20 in WT and mutants. As showed in Supplementary Fig. 4d, the average ratio in *talp-3;ankr-26* double mutant is significantly larger than that in WT. Notably, compared to the *dyf-19*

mutants, the accumulation in *talp-3; ankr-26* double mutants is milder. We reasoned that this could be caused by fewer IFT-B particles entering cilia in double mutants, likely due to the additional function of TALP-3.

2. How do the authors explain their super-resolution microscopy results, where in mammalian cells there does not appear to be a complete co-localization of ANKR26, FBF1 and TALPID3? If they in fact form a complex, the relatively small sizes of the proteins would likely mean that their fluorescent tags should (almost completely) overlap.

Our SIM results showed that the FBF1 and ANKRD26 N-termini are co-localized and both have a mean diameter of ~450 nm. But the mean diameter of the TALPID3 C-terminus (~350 nm) is smaller than those of FBF1 and ANKRD26. SIM images suggest that TALPID3 partially overlaps with FBF1 and ANKRD26. This suggests that the functional interaction among the three proteins likely happens in a restricted region close to the centriolar wall but not along the whole TFs. Supporting evidence is provided by recent stochastic optical reconstruction microscopy (STORM) analysis (ref. 26, Blower et al., *Nature Communications* (2019) 10:993). In this study, the authors reported that ANKRD26 forms a toroid with an inner diameter ~314 nm and an outer diameter of ~578 nm, and FBF1 forms a toroid with an inner diameter ~269 nm and outer diameter of ~496 nm. These available data support the idea that TALPID3, at least partially, colocalizes with FBF1 and ANKRD26.

There is an alternative explanation. Mammalian FBF1, ANKRD26, and TALPID3 are all large proteins (1133, 1710, and 1644 aa, respectively). Also, they are highly coiled-coil proteins that tend to form a “rod”-shape structure. It is very likely that their N-terminus and C-terminus be located far from each other. For example, CEP164, another large TF protein, is organized radially from its C-terminus, the diameter of CEP164 N-terminal signal is very different with that of its C-terminal signal (ref 26, Blower et al. *Nature Communications* (2019) 10:993). In this regard, even the TALPID3 C-terminus does not completely overlap with FBF1 or ANKRD26 N-terminus, the TALPID3 N-terminus may extend and colocalize with FBF1 or ANKRD26.

MINOR ISSUES

1. References in the introduction are dated and therefore less useful for the reader. Many review articles are from 2002-2009, and these miss important new developments (e.g., dealing with ciliopathies).

We appreciate the suggestion and have updated the references.

2. p. 12: obesity is not a 'common' phenotype associated with ciliopathies; it is one of several phenotypes (retinal degeneration, cystic kidneys are more common for example).

We deleted the word “common”.

Reviewer #2:

Minor points:

The ciliary levels of the membrane proteins PKD-2, ODR-10 and OSM-9 in Fig. 2d need to be properly measured and plotted.

As the reviewer suggested, we made those measures and plotted the new data in Fig. 2d.

FBF1 is the only one conserved from *C. elegans* to human and indispensable for TF assembly'. Do the authors mean to say that FBF1 is dispensable to TF assembly?

We do mean that “FBF1 is dispensable for TF assembly”. We are sorry for the typo, and we corrected it in the new version.

Can DYF19 be renamed FBF1? The same question applies to the IFT genes. What purpose does it serve to use the original names when work in every other organism besides nematode uses a universal nomenclature of cilia proteins. This worm exclusivity makes worm studies less accessible to a large audience.

We understand the inconvenience and confusion that this causes for non-worm scientists. This is actually a major issue when preparing a manuscript involving both nematode and mammalian homologs. The nomenclature for *Caenorhabditis elegans* is enforced by the worm community with the rule that “*Gene names must conform to the standard format of 3 or 4 letters gene class name, hyphen, number.*” Also, most gene names have been given based on their mutant phenotype and not their protein function. TALPID3 is more than 4 letters, and so, we have to name it TALP-3. *dyf-19* is the name given to the original mutant retrieved from mutagenesis screen searching for *dye-filling* defect mutant, a functional readout for ciliogenesis. In addition, there is another *C. elegans* gene (Fem-3 mRNA Binding Factor) has been named as FBF-1 many years ago, so we cannot name DYF-19 as FBF-1. For other IFT genes, the names were chosen based on the phenotypes of the mutants long before they were mapped and found to be the homologs of mammalian IFT genes. We simply cannot rename those genes now.

This suggests that TALP-3 and ANKR-26 cooperate to recruit DYF-19 to TFs'. Typo on cooperate

We are sorry for the rushed preparation of the draft. We have carefully proofread the current manuscript.

We first investigated whether TALPID3 and ANKRD26 show any correlation with TFs'. Is correlation the correct word?

We have corrected the typo.

TFs as a whole, independent of the TZ, could constitute a functional gate for both membrane

and soluble proteins.' This idea was previously propose by Ye, Nager and Nachury in a 2018 JCB paper. It has also been covered in various reviews.

Thank the reviewer for the suggestion. We cited the literature accordingly to reflect this fact.

Completely loss of talpid3 disrupts the removal of DCP, a process indirectly blocks the subsequent DA assembly, centriole maturation, and downstream ciliogenesis'. This is not an easy sentence to read. DCP were defined 13 pages before and never mentioned in between. The sentence appears grammatically incorrect, possibly a word missing?

We apologize for the writing. In revised version, we changed the sentence to “Complete loss of TALPID3 disrupts the removal of daughter centriole specific/enriched proteins (DCPs), a prerequisite for DA assembly, centriole maturation, and ciliogenesis.”

Fig. 7a: measurement of intensity need to show individual data points as in 3b.

We measured and revise the figure as suggested by the reviewer.

Reviewer #3:

The following minor issues in the writing need to be addressed;

1) Each sentence in the Introduction needs a reference. The first paragraph on page 5 needs to be fixed.

The references were added, and the first paragraph on page 5 was modified.

2) The term of null should be carefully used. There should be more experiments to support that both alleles are null (e.g. Western or RNA-seq to exclude alternative splicing etc). “putative null” can be safe in many cases without additional experiments.

We appreciate the suggestion and revised the manuscript accordingly to be more accurate.

3) To support “complete colocalization”, quantifications of figure 1 (e.g. line-scan of fluorescence) are required.

We added the line-scan of fluorescence in Supplementary Fig. 3b as suggested by the reviewer.

4) Language can be improved. For example, “When use GFP-tagged IFT-B component OSM-6 as a marker”.

We sincerely apologize for the language issue. The revised version has been proofread by a commercial service.

5) *C.elegans* should be *C. elegans*

We revised the term.

REVIEWERS' COMMENTS:

Reviewer #1 (Remarks to the Author):

Yan and colleagues submitted a revised manuscript aimed at addressing issues I had with the original manuscript. In brief, the major questions were whether the *C. elegans* proteins analysed (TALP-3 and ANKR-26) were 'clearly' homologues/orthologues of the corresponding human proteins (Talpid3 and ANKRD26); whether cilia are present in talp-3;ankr-26 double mutants (in particular).

The authors have gone through great lengths to provide evidence that the two proteins are homologues of their human counterparts. The evidence is largely Blast searches and percentage similarity, which is not unreasonable. But it leaves room for doubt, and I strongly recommend that *alignments* of the proteins with at least the human proteins be included in a Supplementary Figure, so that readers can get a much better sense of the degree of conservation between the proteins.

As far as answering the question regarding the presence/absence of cilia, I am satisfied with the marker (TBB-4) that was used for their experiment.

Other less critical issues were addressed as well.

On the whole, I believe that this is an interesting study, showing an evolutionarily conserved role for three proteins in terms of ciliogenesis/cilium gating. As noted in my original review, the study uses a really nice array of experimental approaches (including super-resolution microscopy, bimolecular fluorescence complementation assays, and biochemistry) in both *C. elegans* and mammalian models to support their model.

The original manuscript was poorly written, but the grammar is now acceptable.

In brief, I am now satisfied with the manuscript, and find that it should be acceptable for publication after the authors provide protein alignments (see above) and fix some of the remaining issues as shown below.

ISSUE:

The PDF I downloaded for the manuscript and figures is large (59 Mb) and I encountered some issues with some of the labelling of axes on graphs. For example, the X axis on Fig. 4, both axes on Fig. 2a. Make sure that all of the Figures/graphs display correctly.

MINOR:

Line 60. change to: "which originate from transformation"

Line 70. the authors write "The fact that only FBF1 among all identified TF components (CEP164, CEP83, CEP89, SCLT1, LRRC45, ANKRD26 and FBF1) is conserved in all ciliated organisms from *C. elegans* to humans" but this paper seems to identify an ANKRD26 homologue in *C. elegans*.

Line 78. TALPID3 is not the official name for the gene, it is KIAA0586. I would at least mention that you 'prefer' to use the deprecated name.

Fig. 4. The graph X axis labels are scrambled (on my computer).

Reviewer #2 (Remarks to the Author):

My minor comments have all been addressed.

I just need to point out that the labeling of the axes in Fig. 2a has gone awry.

Other instance of defective labeling:

categories in Fig. 2c and d

X-axis in Fig. 4a-c

REVIEWERS' COMMENTS:

Reviewer #1 (Remarks to the Author):

Yan and colleagues submitted a revised manuscript aimed at addressing issues I had with the original manuscript. In brief, the major questions were whether the *C. elegans* proteins analysed (TALP-3 and ANKR-26) were 'clearly' homologues/orthologues of the corresponding human proteins (Talpid3 and ANKRD26); whether cilia are present in talp-3;ankr-26 double mutants (in particular).

The authors have gone through great lengths to provide evidence that the two proteins are homologues of their human counterparts. The evidence is largely Blast searches and percentage similarity, which is not unreasonable. But it leaves room for doubt, and **I strongly recommend that *alignments* of the proteins with at least the human proteins be included in a Supplementary Figure**, so that readers can get a much better sense of the degree of conservation between the proteins.

As far as answering the question regarding the presence/absence of cilia, I am satisfied with the marker (TBB-4) that was used for their experiment.

Other less critical issues were addressed as well.

On the whole, I believe that this is an interesting study, showing an evolutionarily conserved role for three proteins in terms of ciliogenesis/cilium gating. As noted in my original review, the study uses a really nice array of experimental approaches (including super-resolution microscopy, bimolecular fluorescence complementation assays, and biochemistry) in both *C. elegans* and mammalian models to support their model.

The original manuscript was poorly written, but the grammar is now acceptable.

In brief, I am now satisfied with the manuscript, and find that it should be acceptable for publication after the authors provide protein alignments (see above) and fix some of the remaining issues as shown below.

Answer:

We thank Reviewer #1 insightful suggestions. We completely agree with the suggestion that an alignment would be helpful for the readers. As suggested, we added the sequence alignment among worm TALP-3 with the N-terminus part of mouse and human TALPID3 in Supplementary Figure 2d. The sequence alignments among worm ANKR-26 and its mouse and human homologs are shown in Supplementary Figure 1.

ISSUE:

The PDF I downloaded for the manuscript and figures is large (59 Mb) and I encountered some issues with some of the labelling of axes on graphs. For example, the X axis on Fig. 4, both axes

on Fig. 2a. Make sure that all of the Figures/graphs display correctly.

We are sorry for the issues. We uploaded our original .psd files multiple times and it appears the conversion between .psd and .pdf file format somehow cause the problem. We thus convert text labels to image layer and now all issues are solved.

MINOR:

Line 60. change to: "which originate from transformation"

We revised it.

Line 70. the authors write "The fact that only FBF1 among all identified TF components (CEP164, CEP83, CEP89, SCLT1, LRRC45, ANKRD26 and FBF1) is conserved in all ciliated organisms from *C. elegans* to humans" but this paper seems to identify an ANKRD26 homologue in *C. elegans*.

Our original meaning is that, before our current work, FBF1 is the only one among identified TF components conserved. Thank for the reviewer suggestion and we agree that this sentence could be misleading. We thus change the sentence to "The fact that among the first 6 TF components (CEP164, CEP83, CEP89, SCLT1, FBF1 and LRRC45) identified, FBF1 is the only one conserved from *C. elegans* to humans".

Line 78. TALPID3 is not the official name for the gene, it is KIAA0586. I would at least mention that you 'prefer' to use the deprecated name.

In the revised version, we use KIAA0586 first, and then, for easily accessible to a large audience, we use TALPID3.

Fig. 4. The graph X axis labels are scrambled (on my computer).

We fixed it.

Reviewer #2 (Remarks to the Author):

My minor comments have all been addressed.

I just need to point out that the labeling of the axes in Fig. 2a has gone awry.

Other instance of defective labeling:

categories in Fig. 2c and d

X-axis in Fig. 4a-c

We are sorry for this error. It was caused by conversion from .psd file to .pdf file when the server automatically converts the file format. We figured out and fixed the labelling problems.